# Polycomb-mediated repression of paternal chromosomes maintains haploid dosage in diploid embryos of *Marchantia*

**Sean Akira Montgomery[1,2]\*, Tetsuya Hisanaga[1], Nan Wang[3], Elin Axelsson[1], Svetlana Akimcheva[1], Milos Sramek[1], Chang Liu[3], Frédéric Berger[1]\***

[1]Gregor Mendel Institute, Austrian Academy of Sciences, Vienna BioCenter, Vienna, Austria; [2]Vienna BioCenter PhD Program, Doctoral School of the University of Vienna and Medical University of Vienna, Vienna, Austria; [3]Institute of Biology, University of Hohenheim, Stuttgart, Germany

**Abstract:** Complex mechanisms regulate gene dosage throughout eukaryotic life cycles. Mechanisms controlling gene dosage have been extensively studied in animals, however it is unknown how generalizable these mechanisms are to diverse eukaryotes. Here, we use the haploid plant *Marchantia polymorpha* to assess gene dosage control in its short-lived diploid embryo. We show that throughout embryogenesis, paternal chromosomes are repressed resulting in functional haploidy. The paternal genome is targeted for genomic imprinting by the Polycomb mark H3K27me3 starting at fertilization, rendering the maternal genome in control of embryogenesis. Maintaining haploid gene dosage by this new form of imprinting is essential for embryonic development. Our findings illustrate how haploid-dominant species can regulate gene dosage through paternal chromosome inactivation and initiates the exploration of the link between life cycle history and gene dosage in a broader range of organisms.

## Editor's evaluation

Mechanisms for controlling gene dosage and uniparental gene expression vary widely across the eukaryotic tree, with many such mechanisms still unknown. This article describes an epigenetic mechanism used to modulate paternal chromosome gene dosage during the transient diploid state of the primarily haploid plant, Marchantia polymorpha. This fascinating case of genome-wide genomic imprinting will be of broad interest to scientists studying gene expression, early development, and especially those focused on understanding the diversity of gene dosage control mechanisms.

## Introduction

Maintaining proper gene dosage is a challenge for eukaryotic organisms. For instance, multi-subunit protein complexes require balanced production of each component, lest incomplete, non-functional complexes are produced (*Birchler and Veitia, 2010*). Misregulation of gene dosage can lead to developmental defects, sterility, and disease (*Loda et al., 2022*). Dramatic changes in gene dosage notably occur during the process of diploidization after whole genome duplication (*Edger and Pires, 2009*) and sex chromosome evolution (*Mank, 2013*). Sex chromosome dosage compensation is best understood mechanistically in mammalian female X chromosome inactivation (XCI) (*Żylicz and Heard, 2020*) and *Drosophila* male X chromosome upregulation (*Samata and Akhtar, 2018*). However, the molecular mechanisms are not conserved across the many diploid-dominant species in which sex

**\*For correspondence:**
sean.montgomery@gmi.oeaw.ac.at (SAM);
Frederic.berger@gmi.oeaw.ac.at (FB)

**Competing interest:** The authors declare that no competing interests exist.

**eLife digest** The reproductive cells of organisms that reproduce sexually – the egg and the sperm – each contain one copy of the organism's genome. An embryo forms upon fertilization of an egg by a sperm cell. This embryo contains two copies of the genome, one from each parent. Under most circumstances, it does not matter which parent a gene copy came from: both gene copies are expressed. However, in some species genes coming from only one of the parents are switched on. This unusual mode of gene expression is called genomic imprinting. The best-known example of this occurs in female mammals, which repress the genes on the paternal X chromosome. Genomic imprinting also exists in flowering plants.

Both mammals and flowering plants evolved tissues that channel nutrients from the mother to the embryo during development; the placenta and the endosperm, respectively. Genomic imprinting had, until now, only been described in these two types of organisms. It was unknown whether imprinting also happens in other organisms, and specifically those in which embryos develop inside the mother but without the help of a placenta or endosperm.

Here Montgomery et al. addressed this question by studying the liverwort, *Marchantia polymorpha*, a moss-like plant. Initial experiments showed that cells in the liverwort embryo mostly expressed the genes coming from the egg, and not the sperm. All the genetic material coming from the sperm had a molecular marker or tag called H3K27me3. This mark, which also appears on the paternal X chromosome in female mammals, switches off the genes it tags. *M. polymorpha* embryos thus suppress gene expression from all of the genetic material from the father, relying only on maternal genetic material for development. When Montgomery et al. deleted the maternal genes necessary for making the H3K27me3 mark, the paternal genes switched on, and this led to the death of the embryos. The survival of *M. polymorpha* embryos therefore depended on keeping only one set of genes active.

Taken together these experiments indicate that genomic imprinting evolved about 480 million years ago, about 320 million years earlier than previously thought, in organisms for which embryo development depended only on one parent. This means there are likely many more organisms that control gene expression in this way, opening up opportunities for further research. Understanding imprinting in more detail will also shed light on how sexual reproduction evolved.

chromosome dosage compensation has been described (*Gu et al., 2019*; *Gu and Walters, 2017*; *Lau and Csankovszki, 2015*; *Lucchesi and Kuroda, 2015*; *Muyle et al., 2012*) potentially due to the repeated innovation of sex chromosomes (*Bachtrog et al., 2014*). Gene dosage also changes regularly during cell cycles and life cycles as ploidy levels change. Therefore, a large variety of gene dosage regulatory mechanisms remain to be discovered in eukaryotes.

All sexually reproducing eukaryotes have diploid and haploid life cycle stages, but the duration of each stage varies greatly amongst species. The alternation between ploidy must be programmed because unscheduled change in ploidy leads to genome instability (*Davoli and de Lange, 2011*). Despite the short haploid stage of gametes in mammals, gene dosage is managed by meiotic sex chromosome inactivation and post-meiotic silencing in male gametes (*Lee and Bartolomei, 2013*; *Namekawa et al., 2006*). This is continued as imprinted XCI in early female embryos, wherein the male X chromosome is selectively repressed (*Takagi and Sasaki, 1975*). The disruption of meiotic sex chromosome inactivation results in meiotic arrest, illustrating its essentiality for sexual reproduction (*Turner, 2007*). However, the mechanisms of gene dosage control throughout the mammalian life cycle are not conserved amongst animals (*Maine, 2010*; *Turner, 2015*; *Vibranovski, 2014*), reflective of the diversity of sex chromosome dosage compensation mechanisms (*Gu et al., 2019*; *Gu and Walters, 2017*; *Lau and Csankovszki, 2015*; *Lucchesi and Kuroda, 2015*; *Muyle et al., 2012*). Most eukaryotic life cycles differ from that of animals, with a predominance of haploid life stages, suggesting that there may be extensive diversity yet uncovered. Haploid and haploid-dominant species present an intriguing and understudied corollary to understand gene dosage control throughout life cycles. Strictly haploid species such as yeast show limited evidence for gene dosage control (*Chen et al., 2020*; *Hose et al., 2015*; *Springer et al., 2010*). Haploid-dominant species with a short diploid phase of development are of particular interest because of the stark contrast of life cycles with diploid-dominant species and their prevalence across various branches of eukaryotic

life. How, or even if, haploid-dominant species balance gene dosage during the diploid phase is not known.

Here, we identified a control of gene dosage by selective repression of alleles of paternal origin in the diploid embryonic stage of the model haploid-dominant bryophyte *Marchantia polymorpha* (hereafter referred to as *Marchantia*). We show that *Marchantia* represses paternal chromosomes by genomic imprinting via the Polycomb mark H3K27me3. This unique form of genomic imprinting, which we term 'paternal chromosome repression' (PCR), is the first description of imprinting in the bryophyte lineage since its theoretical prediction (*Carey et al., 2021*; *Haig, 2013*; *Haig and Wilczek, 2006*; *Montgomery and Berger, 2021*; *Shaw et al., 2011*). Disruption of PCR results in derepression of the paternal genome and lethality. Furthermore, we show that the imprinting mark is deposited at the pronuclear stage and initiates PCR that persists until the end of embryogenesis. Therefore, *Marchantia* manages gene dosage by effectively maintaining a functionally haploid state in diploid embryos under the control of the maternal genome.

## Results

### Embryonic transcription is maternally biased

To explore *Marchantia* gene dosage control, we performed crosses between two wild-type natural accessions, Cam-2 as the mother and Tak-1 as the father, and obtained transcriptomes from embryos 13 days after fertilization (daf) (*Figure 1A*). Each transcriptome was prepared from single hand-dissected embryos that were washed several times to remove potential contaminating RNA from the surrounding maternal tissue (*Figure 1B*; *Video 1*; *Schon and Nodine, 2017*). A comparison of embryonic, vegetative, and sexual organ transcriptomes demonstrated the distinctness of embryonic transcriptomes from other tissues and the similarity of each embryonic transcriptome to each other (*Figure 1C* and *Figure 1—figure supplement 1A*). Together, these results indicated to us that we had obtained pure embryonic transcriptomes for further analyses.

We further looked for evidence of allele-specific expression in diploid embryos. We utilized single-nucleotide polymorphisms (SNPs) between female and male accessions to calculate the ratio of reads originating from maternal alleles versus paternal alleles (maternal ratio, $p_m$; ascribed a value between 0 and 1, ranging from 0 if only paternal reads were detected to 1 if only maternal reads were detected). Combining all replicates, we only considered genes with at least 50 reads containing informative SNPs. Transcription was overall maternally biased for 98% of resolved genes, with 73% of genes maternally biased (as defined in *Wang and Clark, 2014*; $0.65 \le p_m < 0.95$) and 25% of genes only expressed from maternal alleles ($p_m \ge 0.95$) (*Figure 1D–E*). The strong unidirectional bias in gene expression suggested homogeneity amongst replicates, which was confirmed when assessing the maternal ratio of transcription from each replicate (*Figure 1—figure supplement 1B*). We conclude that in *Marchantia* embryos, genes are primarily or exclusively expressed from their maternal allele.

The exact reciprocal cross was not possible because inbred genetically near-identical pairs of males and females do not exist, but to confirm that the maternal bias did not result from the pair of natural accessions used, we analyzed published RNA-seq data from a cross of different accessions, Tak-2 and Tak-1 (*Frank and Scanlon, 2015*). The published transcriptomes were generated from samples collected by laser-capture microdissection, an orthogonal sample collection method that offered equally high sample purity (*Schon and Nodine, 2017*). A similarly strong maternal bias in transcription was observed, with 99% of genes maternally biased or expressed only from maternal alleles (*Figure 1—figure supplement 1C-D*). Thus, these data ruled out that the observed allele-specific gene expression originated from natural variation amongst wild-type parents or from maternal contamination during sample collection. Additionally, we tested whether the genes resolved by our analyses formed a sample representative of all genes. We found no correlation between the maternal ratio and expression level of a gene (*Figure 1—figure supplement 1E*), nor did the transcription maternal ratio vary significantly along the length of each autosome (*Figure 1F* and *Figure 1—figure supplement 1F*). Thus, we inferred that the genes we were able to resolve with SNPs were representative of a genome-wide maternal bias in transcription. Overall, the lack of paternal allele expression suggests the presence of a repressive chromatin modification specifically on the paternal genome.

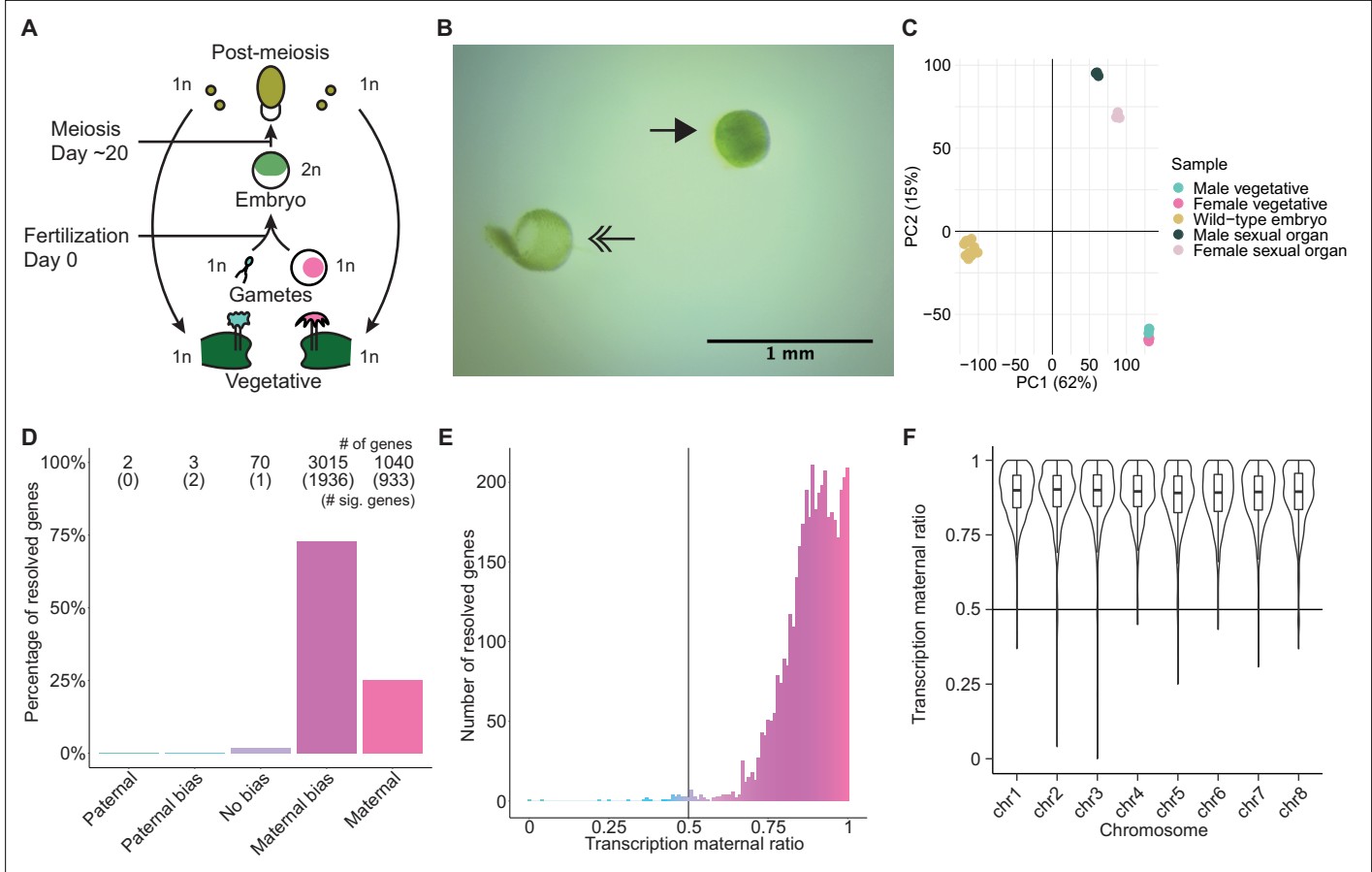

**Figure 1.** Embryonic transcription is maternally biased. (**A**) Life cycle of *Marchantia polymorpha*. Haploid (1n) vegetative males and females produce male and female reproductive structures, which subsequently produce sperm and egg. The diploid (2n) embryo persists for around 20 days before meiosis and the production of haploid spores. Ploidy of each stage is indicated. (**B**) Image of a representative hand-dissected embryo after removal of perianth and calyptra of maternal origin. Solid single arrow indicates isolated embryo. Double arrow indicates the removed calyptra. Scale bar as indicated. (**C**) Principal component analysis of transcriptomes from wild-type embryos (Cam-2 × Tak-1), vegetative tissues from female and male parents, and female and male sexual organs. The first two principal components are plotted, and the percentage of variance explained is indicated. (**D**) Percentage of measured genes within each category of maternal ratio ($p_m$) of transcription in wild-type embryos. Segments are for paternal ($p_m \leq$ 0.05), paternal bias (0.05<$p_m \leq$ 0.35), no bias (0.35 <$p_m$ < 0.65), maternal bias (0.65 ≤$p_m$ < 0.95), and maternal (0.95 ≤ $p_m$) expression of genes, with the number of genes indicated above each bar and the number of genes significantly deviating from $p_m$ = 0.5 in parentheses. Significance was assessed using an exact binomial test with Bonferroni correction. (**E**) Histogram of the maternal ratio ($p_m$) of transcription per gene in wild-type (Cam-2 × Tak-1) embryos. Each bin is 0.01 units wide. (**F**) Violin plots of transcription maternal ratio of genes per chromosome. Sex chromosomes are excluded as alleles could not be resolved.

The online version of this article includes the following figure supplement(s) for figure 1:

**Figure supplement 1.** Maternally biased transcription in embryos.

## Levels of H3K27me3 enrichment are paternally biased

To better understand what chromatin-related mechanisms may be driving the maternal bias in embryonic transcription, we examined differentially expressed genes between vegetative parents and embryos. In total, 3879 genes were upregulated in embryos relative to both mothers and fathers (*Figure 2—figure supplement 1A*), while 3466 genes were downregulated (*Figure 2—figure supplement 1B*). Upregulated genes were more expressed from the maternal genome than downregulated genes (*Figure 2—figure supplement 1C*, effect size [Cohen's *d*]=0.276) highlighting a maternal control over the embryonic transcriptome. Since imprinting is an epigenetic process, we focused further on chromatin-related genes. Of the 215 chromatin-related genes in the *Marchantia* genome (*Bowman et al., 2017*), 151 were upregulated and 7 were downregulated (*Figure 2—figure supplement 1D*; *Supplementary file 1*). Of these, 20 genes were specifically expressed in the embryonic

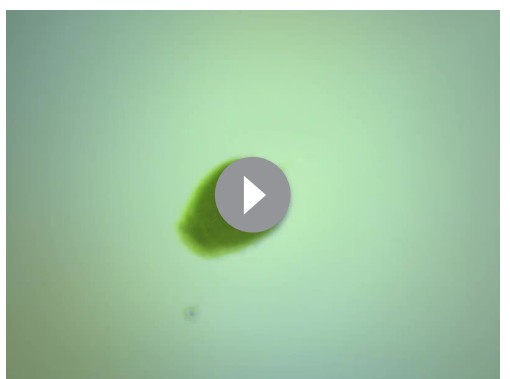

**Video 1.** Movie of the dissection of a representative *Marchantia* embryo from surrounding calyptra of maternal origin.

https://elifesciences.org/articles/79258/figures#video1

stage (*Supplementary file 1*, transcripts per million [TPM] greater than 1 in embryos and less than 1 in other tissues). Two noteworthy genes were paralogs of the catalytic subunit of the Polycomb repressive complex 2 (PRC2), *E(z)2* and *E(z)3* (*Figure 2A*). PRC2 is a conserved multi-subunit complex that deposits H3K27me3 and is associated with gene silencing (*Margueron and Reinberg, 2011*). The genes encoding the other three subunits of PRC2, *FIE*, *Su(z)12*, and *MSI1*, and a third catalytic subunit paralog, *E(z)1*, were expressed in all tissues (*Figure 2A*). Therefore, we hypothesized that H3K27me3 might be present on silenced paternal alleles in the *Marchantia* embryo.

We first set out to determine whether H3K27me3 was enriched on paternal alleles in *Marchantia* embryos. We profiled chromatin modifications using CUT&RUN (*Skene and Henikoff, 2017*; *Zheng and Gehring, 2019*) on sorted nuclei from *Marchantia* embryos. We used SNPs between male and female accessions to distinguish the parental allele of origin for CUT&RUN reads to calculate a maternal ratio ($p_m$) for genomic regions of interest. Enrichment in H3K27me3 was paternally biased for 88% of genes resolved ($0.05 < p_m \leq 0.35$) (*Figure 2B–C*) and genes with paternally biased H3K27me3 had maternally biased transcription (*Figure 2D*). Fewer genes were resolved as statistically significant compared to transcriptomic analyses, as assessed using an exact binomial test with Bonferroni correction, due to fewer CUT&RUN experiment replicates being performed, as is usual for chromatin profiling experiments. Genes paternally marked with H3K27me3 were located across all autosomes (*Figure 2E* and *Figure 2—figure supplement 1E*), indicating the broad, pervasive nature of the phenomenon. In contrast, a paternal bias was not observed in profiles of H3K9me1, H3, and H3K36me3 (5%, 2%, and 2% of genes with $0.05 < p_m \leq 0.35$, respectively) (*Figure 2—figure supplement 1F-H*). We conclude that levels of H3K27me3 enrichment anticorrelate with maternally biased transcription and spreads over most paternal alleles in *Marchantia* embryos. These findings suggest that H3K27me3 covers the entire genome of paternal origin.

## Partitioning of the paternal genome into dense H3K27me3 compartments

To test if the paternal genome was coated with H3K27me3, we performed immunofluorescence experiments to observe the localization of this modification within embryonic nuclei. As a control, nuclei of parental adult vegetative cells showed evenly distributed speckles of heterochromatin marked by H3K27me3 (*Figure 3A*; *Montgomery et al., 2020*). In stark contrast, one to three large heterochromatic compartments, as defined by dense DNA staining, covered 10% of the area in embryonic nuclei (*Figure 3A* and *Figure 3—figure supplement 1A-B*). A strong correlation between heterochromatic foci and H3K27me3 was apparent (*Figure 3A*); 44% of the H3K27me3 signal was contained within these compartments (*Figure 3—figure supplement 1C*), whereas 70% of the area of the compartments was contained within H3K27me3 domains (*Figure 3—figure supplement 1D*). These heterochromatic domains associated with H3K27me3 were also apparent in a cross with a different male accession (Cam-1), suggesting that the imprinted status in the embryo does not depend solely on Tak-1 (*Figure 3—figure supplement 1E*). In contrast, only 20% and 10% of H3K9me1 and H3K36me3 signal, respectively, were contained within heterochromatic compartments (*Figure 3—figure supplement 1C*). H3K9me1 is indicative of constitutive heterochromatin on repetitive genomic regions in Marchantia, whereas H3K36me3 is associated with expressed genes (*Montgomery et al., 2020*). We conclude that the portion of the genome marked by H3K27me3 represents the largest fraction of heterochromatin organized as a couple of dense compartments in embryonic nuclei.

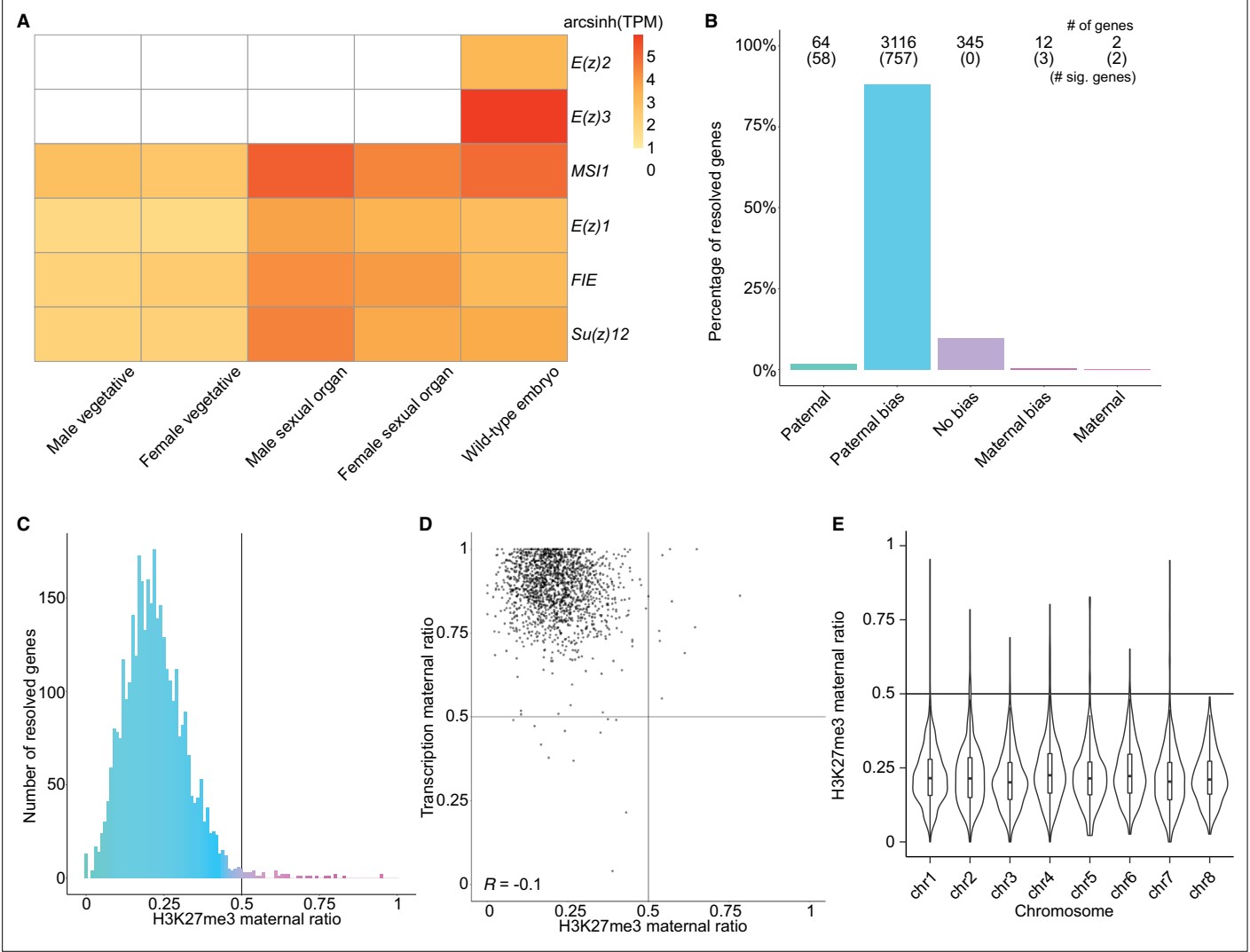

**Figure 2.** Levels of H3K27me3 enrichment are paternally biased. (**A**) Heatmap of gene expression of Polycomb repressive complex 2 subunits across *Marchantia* development. Vegetative male (Tak-1) and female (Cam-2) tissues give rise to male and female sexual organs (antheridiophores and archegoniophores, respectively; data from *Higo et al., 2016*). Wild-type embryos are from Cam-2 × Tak-1 crosses. Values shown are arcsinh transformed transcript per million values. (**B**) Percentage of measured genes within each category of maternal ratio ($p_m$) of H3K27me3 in wild-type embryos. Segments are for paternal ($p_m \leq 0.05$), paternal bias ($0.05 < p_m \leq 0.35$), no bias ($0.35 < p_m < 0.65$), maternal bias ($0.65 \leq p_m < 0.95$), and maternal ($0.95 \leq p_m$) H3K27me3 of genes, with the number of genes indicated above each bar and the number of genes significantly deviating from $p_m = 0.5$ in parentheses. Significance was assessed using an exact binomial test with Bonferroni correction. (**C**) Histogram of the maternal ratio ($p_m$) of H3K27me3 per gene in wild-type (Cam-2 × Tak-1) embryos. Each bin is 0.01 units wide. (**D**) Scatterplot of maternal ratios of H3K27me3 and transcription per resolved gene. Spearman correlation is indicated. (**E**) Violin plots of H3K27me3 maternal ratio of genes per chromosome in wild-type embryos. Sex chromosomes are excluded as alleles could not be resolved.

The online version of this article includes the following figure supplement(s) for figure 2:

**Figure supplement 1.** Paternally biased H3K27me3 in embryos.

## All paternal autosomes are coated with H3K27me3

The *Marchantia* life cycle has a vegetative haploid phase with a diploid phase of embryogenesis (*Figure 1A*). In *Marchantia* diploid embryonic cells, the genome is packaged in 2*n*=18 chromosomes including two sex chromosomes and sixteen autosomes. The large size of the heterochromatic compartments marked by H3K27me3 suggested that they contained entire chromosomes (*Figure 3A*). Accordingly, immunofluorescence of mitotic cells revealed that eight of the sixteen autosomes were densely coated with H3K27me3, whereas the other half had no detectable H3K27me3 (*Figure 3B*). In contrast, mitotic vegetative haploid cells showed an uneven speckled pattern of H3K27me3 over

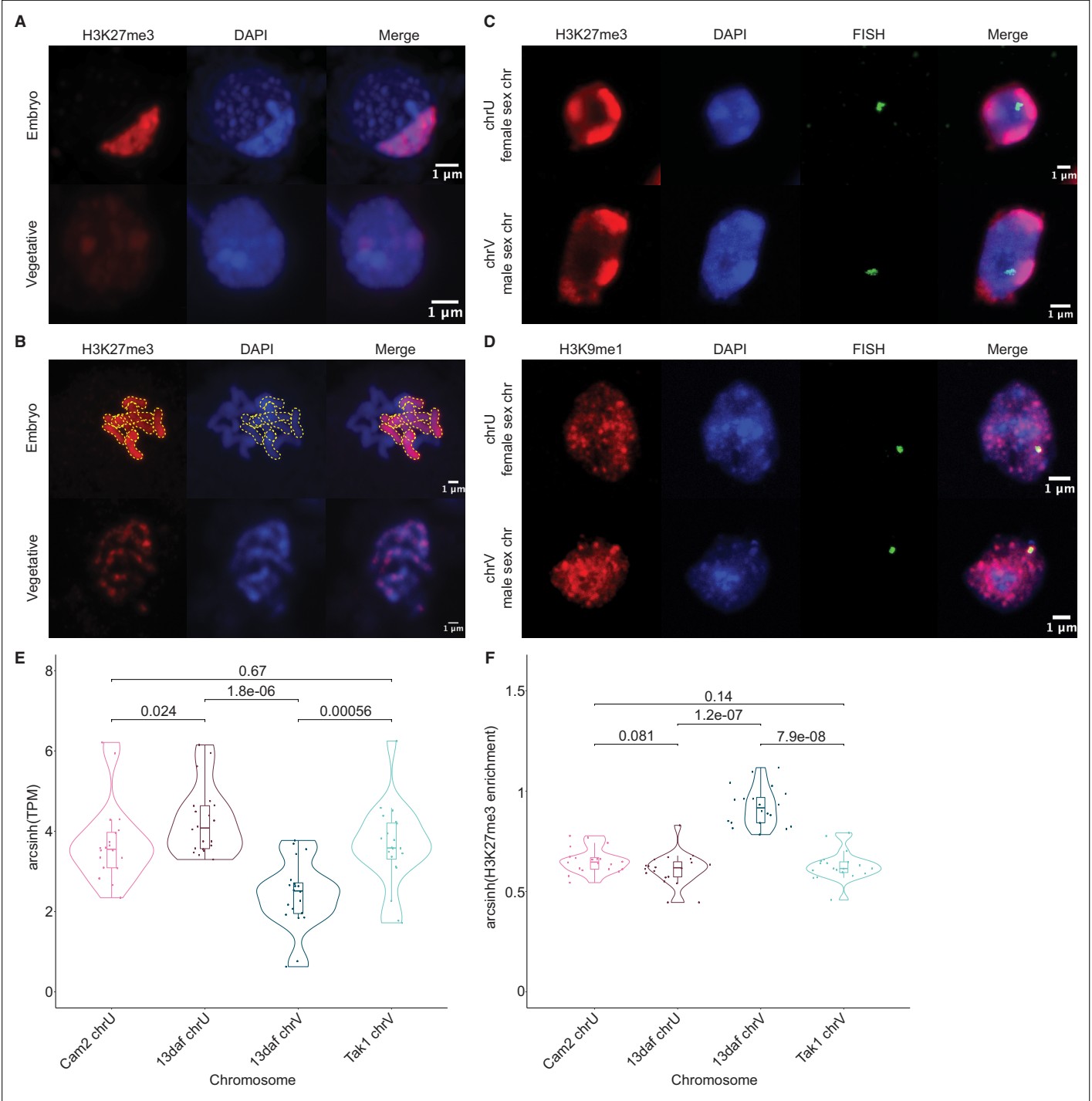

**Figure 3.** Paternal autosomes are coated in H3K27me3 and partitioned in heterochromatic foci. (**A**) Immunofluorescence of H3K27me3 in interphase wild-type embryonic and vegetative nuclei. DNA is stained with DAPI. Scale bar as indicated. (**B**) Immunofluorescence of H3K27me3 in mitotic wild-type embryonic and vegetative cells. DNA is stained with DAPI. Contrast was enhanced for the DAPI channel of vegetative nuclei for illustration purposes. Outlines of the H3K27me3-coated chromosomes are indicated with dashed yellow lines. Scale bar as indicated. (**C**) Immuno-FISH for sex chromosomes and H3K27me3 in interphase wild-type embryonic nuclei. The female sex chromosome is chrU and the male sex chromosome is chrV. Scale bar as indicated. (**D**) Immuno-FISH for sex chromosomes and H3K9me1 in interphase wild-type embryonic nuclei. The female sex chromosome is chrU and the male sex chromosome is chrV. Scale bar as indicated. (**E**) Violin plot of arcsinh transformed transcript per million (TPM) values for sex chromosome gametologs in vegetative (Cam2 and Tak1) and embryonic (13 days after fertilization [daf]) samples. p-Values are indicated, unpaired two-tailed Wilcoxon test. (**F**) Violin plot of arcsinh transformed H3K27me3 enrichment for sex chromosome gametologs in vegetative and embryonic samples. p-Values are indicated, unpaired two-tailed Wilcoxon test.

*Figure 3 continued on next page*

*Figure 3 continued*

The online version of this article includes the following figure supplement(s) for figure 3:

**Figure supplement 1.** Quantification of immunofluorescence experiments.

the eight autosomes (*Figure 3B*). This observation mirrored the strong paternal bias of H3K27me3 enrichment. We conclude that the paternal genome is covered by H3K27me3 and partitioned into heterochromatic compartments within embryonic nuclei.

In addition to the eight autosomes, each parent carries a small sex chromosome (U in females and V in males; *Iwasaki et al., 2021*). Sex chromosomes detected using fluorescence in situ hybridization (FISH) were not associated with H3K27me3 heterochromatic foci in *Marchantia* embryos (*Figure 3C* and *Figure 3—figure supplement 1E*). Instead, we observed that both U and V sex chromosomes rather associated with H3K9me1 heterochromatic foci (*Figure 3D* and *Figure 3—figure supplement 1E*). We conclude that the sex chromosomes are excluded from H3K27me3 heterochromatic compartments and form small constitutive heterochromatic foci in both embryonic and vegetative nuclei. Yet, the protein coding genes on the female U sex chromosome are expressed at a much higher level than homologous genes on the male V sex chromosome (*Figure 3E*). This imbalance towards female expression is correlated with an enrichment of H3K27me3 on the genes of the male chromosome (*Figure 3F* and *Figure 3—figure supplement 1F*). Hence, overall, H3K27me3 targets the paternal alleles of all chromosomes in *Marchantia*, resulting in a pseudohaploid state in the embryo.

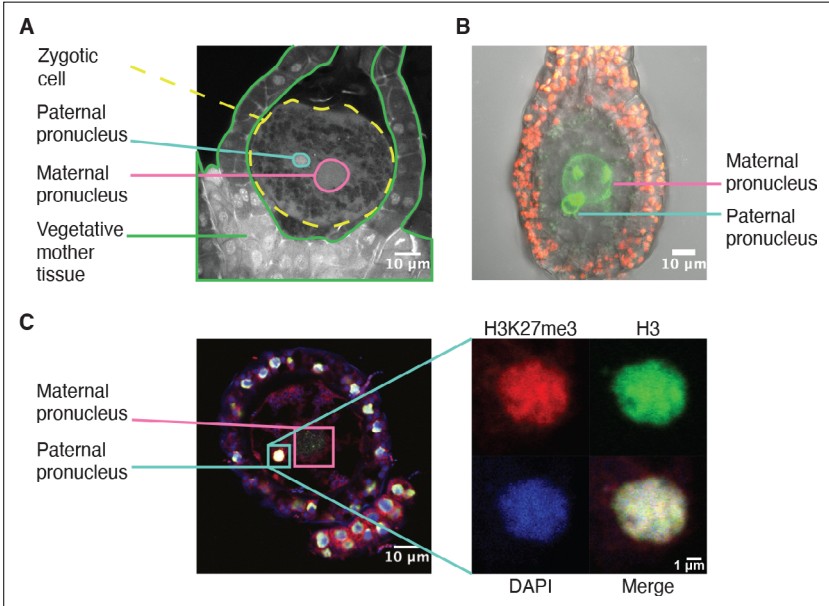

**Figure 4.** H3K27me3 is deposited in paternal pronuclei. (**A**) Annotated confocal image of a *Marchantia* zygote 3 days after fertilization (daf) with surrounding vegetative mother tissue. The paternal pronucleus is visible in the vicinity of the maternal pronucleus. Nuclei are stained with DAPI. Indicated are the fertilized zygotic cell (dashed yellow circle), maternal pronucleus (pink circle), vegetative mother tissue (green lines) surrounding the zygote, and paternal pronucleus (cyan circle). Scale bar as indicated. (**B**) Composite maximum intensity projection confocal image of a *Marchantia* zygote expressing *SUN-GFP* at 3 daf plus surrounding vegetative mother tissue. Nuclear membranes are marked by localization of SUN-GFP, shown in green. The paternal pronucleus is smaller than and adjacent to the maternal pronucleus. Autofluorescence from chloroplasts in vegetative mother cells is shown in red, and both channels are overlayed on a transmitted light image. Scale bar as indicated. (**C**) Immunofluorescence image 3 daf of a *Marchantia* zygote. Both maternal and paternal pronuclei are indicated in pink and cyan, respectively. The inset depicts a zoomed-in view of the paternal pronucleus with separate images for H3K27me3 (red), H3 (green), DAPI (blue), and the merged image. Contrast is enhanced for each image and channel independently for visualization purposes. Scale bars as indicated.

The online version of this article includes the following figure supplement(s) for figure 4:

**Figure supplement 1.** Immunofluorescence of pronuclei.

# H3K27me3 is deposited in the male pronucleus

Like in most animals, the male pronucleus contributed by the sperm remains separated from the female pronucleus contributed by the egg in the *Marchantia* zygote, thus providing an opportunity for the deposition of an epigenetic mark on the genome of one parent (*Hisanaga et al., 2021*; *Hisanaga et al., 2019*). As *Marchantia* sperm chromatin is comprised of protamines and is devoid of histones (*D'Ippolito et al., 2019*; *Reynolds and Wolfe, 1978*), we hypothesized that paternal H3K27me3 is deposited on paternal alleles sometime after fertilization. Male and female pronuclei remain separate until 4 daf (*Figure 4A–B*; *Hisanaga et al., 2021*), thus we examined if H3K27me3 was deposited before pronuclear fusion. As we could not isolate pronuclei for chromatin profiling, we instead performed immunofluorescence experiments. At 2 daf, we observed H3 in the paternal pronucleus, but did not observe H3K27me3 (*Figure 4—figure supplement 1A*). At 3 daf, we observed both H3K27me3 and H3 in the paternal pronucleus (*Figure 4C* and *Figure 4—figure supplement 1*), demonstrating that H3K27me3 is deposited in the paternal pronucleus before its fusion with the maternal pronucleus. The delay between the detection of H3 and H3K27me3 suggests that the mark is deposited after nucleosomes are assembled, but we cannot exclude that a level of H3K27me3 below the limit of detection is present at 2 daf. The timing of H3K27me3 deposition is not unique, as low levels of H3K9me1 also appear in paternal pronuclei at 3 daf (*Figure 4—figure supplement 1B*). We do not make any comparisons to the maternal pronucleus as its large size made it difficult to capture in a single paraffin slice. Therefore, our data suggest that paternal alleles become imprinted by H3K27me3 while they are

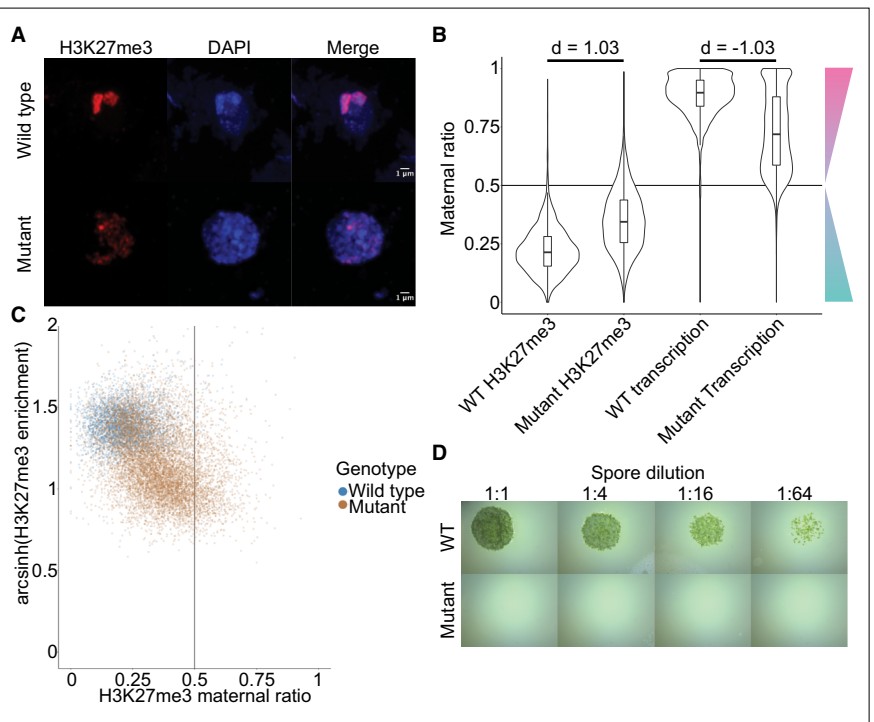

**Figure 5.** Embryonic Polycomb repressive complex 2 (PRC2) deposits H3K27me3 and represses the paternal genome. (**A**) Immunofluorescence of H3K27me3 in interphase wild-type (WT) and mutant embryonic nuclei. DNA is stained with DAPI. (**B**) Violin plots of maternal ratios for WT and mutant H3K27me3 and transcription. Cohen's *d* effect size values are indicated for pairwise comparisons of mutant to WT H3K27me3 maternal ratio and mutant to WT transcription maternal ratio, where |*d*| > 0.8 is a large effect, as previously reported (*Cohen, 1992*). (**C**) Scatterplot of H3K27me3 enrichment versus H3K27me3 maternal ratio per gene in WT and mutant embryos. Genes with an arcsinh-transformed H3K27me3 enrichment greater than 2 are displayed as triangles at the upper boundary of the plot. (**D**) Spore germination assay for spores resulting from WT and mutant embryos. A serial dilution of a suspension of spores from a single embryo is shown.

The online version of this article includes the following figure supplement(s) for figure 5:

**Figure supplement 1.** Chromatin phenotypes of e(z)2/e(z)3 mutants.

**Figure supplement 2.** Transcription phenotypes in mutant embryos.

spatially segregated from maternal alleles prior to the fusion of pronuclei. The conservative restoration of H3K27me3 after DNA replication (*Jiang and Berger, 2017*) provides a mechanism to propagate the initial 'coat' of H3K27me3 to paternal chromosomes and silence the paternal genome throughout embryonic development.

## Embryonic PRC2 deposits H3K27me3 and represses the paternal genome

To directly test the effect of embryo-specific PRC2 subunits on paternal H3K27me3 imprinting, we knocked out both embryo-specific paralogs of the catalytic subunit, *E(z)2* and *E(z)3* (*Figure 5—figure supplement 1A*). These mutants did not display any aberrant phenotype prior to fertilization (*Figure 5—figure supplement 1B, C*). We crossed mutant females (Cam-2 *e(z)2/e(z)3*) to wild-type males and observed a loss or dispersion of large heterochromatic foci that overlapped with H3K27me3 foci in embryos compared to wild-type embryos (*Figure 5A* and *Figure 5—figure supplement 1D*). In these embryos, enrichment of H3K27me3 as measured by CUT&RUN was significantly decreased over most genes (*Figure 5—figure supplement 1E*, Wilcoxon signed-rank test p < 0.0001). Thus, maternal inheritance of both *e(z)2* and *e(z)3* altered patterns of H3K27me3-associated heterochromatic foci and reduced H3K27me3 deposition. The paralog *E(z)1* is expressed in embryos (*Figure 2A*) and likely accounted for the remaining detected H3K27me3, but it proved impossible to test this hypothesis due to lethality at the haploid vegetative stage in knockdowns of *E(z)1* (*Flores-Sandoval et al., 2016*). It is possible that some of the remaining H3K27me3 is also due to expression of paternal *E(z)2* and *E(z)3*. To determine whether paternal alleles were the source of H3K27me3 loss, we distinguished the parental genome of origin of CUT&RUN sequencing reads and calculated maternal ratios. The paternally biased enrichment of H3K27me3 was significantly reduced (*Figure 5B*, Wilcoxon signed-rank test p < 0.0001, effect size = 1.03) and only 51% of genes were categorized as paternally biased ($0.05 < p_m \leq 0.35$) (*Figure 5—figure supplement 1F*), down from 88% in wild type (*Figure 2B*). Furthermore, there was a negative correlation between H3K27me3 enrichment and maternal ratio (*Figure 5C*), indicating that loci that lost H3K27me3 had predominantly lost paternal H3K27me3 in embryos that lost the maternal alleles of *E(z)2* and *E(z)3*. We conclude that the deposition of H3K27me3 on most paternal loci in *Marchantia* embryos depends on maternally supplied PRC2 activity.

If H3K27me3 did indeed repress paternal alleles, we expected expression from paternal alleles at loci that lost paternal H3K27me3 in *e(z)2/e(z)3* mutants. To test this idea, we generated transcriptomes from mutant embryos and examined maternal ratios of gene expression. Overall, transcription became more biallelic in mutants than in wild-type embryos (*Figure 5B*, Wilcoxon signed-rank test p < 0.0001, effect size = –1.03). Only 47% of genes were maternally biased ($0.65 \leq p_m < 0.95$) and 15% of genes completely expressed from maternal alleles ($p_m \geq 0.95$), a stark deviation from wild-type values of 73% and 25% (*Figure 5—figure supplement 2A* and compare with *Figure 1D*). Comparing both mutant maternal ratios of transcription with patterns of H3K27me3, we observed that the H3K27me3 maternal ratio negatively correlated with the maternal ratio of transcription (*Figure 5—figure supplement 2B-C*) and that H3K27me3 enrichment positively correlated with the maternal ratio of transcription (*Figure 5—figure supplement 2B, D*), indicating that loci with less paternal H3K27me3 in mutants were more transcribed from paternal alleles. High paternal H3K27me3 and maternal transcription did not correlate with the level of gene expression (*Figure 5—figure supplement 2B, E-G*), suggesting that absolute gene expression levels did not influence paternal allele repression. Therefore, paternal alleles regain expression in the absence of maternal PRC2 and upon the loss of paternal H3K27me3. A differential gene expression analysis found 3824 and 2003 genes were upregulated and downregulated, respectively, in mutant embryos. There was a significant overlap between downregulated genes and genes overlapping a peak of H3K27me3 in mutant embryos ($\chi^2$ test, p-value = 3.616e-15, *Supplementary file 2*), suggesting that downregulated genes were repressed by residual PRC2 activity. Yet the large majority of misregulated genes were upregulated and we conclude that H3K27me3 deposited by embryo-specific PRC2 subunits is required for the collective repression of paternal alleles, resulting in PCR.

To assess the physiological relevance of the loss of paternal allele repression, we quantified the growth and survival of mutant embryos. Embryonic growth was significantly slower in mutants than in wild type, as measured by total size (*Figure 5—figure supplement 2H*). Only 20% of mutant embryos survived to maturity versus 95% of wild type (*Figure 5—figure supplement 2I*), and only 5% of all

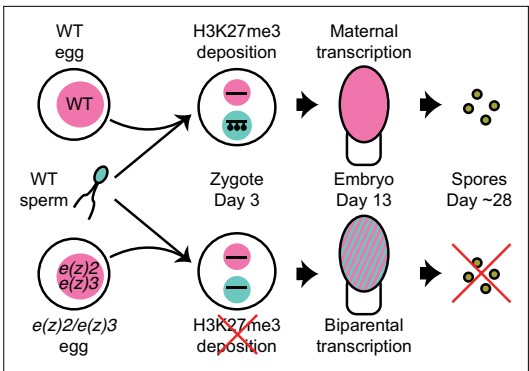

**Figure 6.** Model of genomic imprinting in *Marchantia*. Model of H3K27me3 deposition in wild-type (WT) paternal pronuclei and subsequent propagation throughout embryogenesis. Closed lollipops depict H3K27me3 on genes. Pink and blue circles depict maternal and paternal (pro)nuclei, respectively. Pink and striped ovals depict whole embryos and the parental genome from which transcription is occurring. Yellow discs depict mature spores. The lack of H3K27me3 on paternal pronuclei in mutant zygotes allows for the transcription of paternal alleles in the embryo, ultimately leading to the lack of viable spore production.

mutant embryos produced spore-bearing structures, compared to 77% of wild type (*Figure 5—figure supplement 2J*). Of those mutants that produced spores, none produced viable spores, thus rendering all mutants unable to continue their life cycle (*Figure 5D*). Although it is not possible to exclude that ectopic expression of genes due to the loss of PRC2 activity but not the loss of imprinting is the direct cause of the loss of viability, our results are also consistent with the model that PRC2-mediated PCR is essential for viability and fecundity in *Marchantia* (*Figure 6*).

## Discussion

In the present study, we report how a haploid-dominant species controls gene dosage during a short diploid stage. While diploid-dominant species often balance gene dosage between autosomes and sex chromosomes throughout their life cycles, haploid-dominant species must manage gene dosage only during embryonic development. The model bryophyte *M. polymorpha* achieves gene dosage control during embryonic development via genomic imprinting and subsequent repression of the paternal genome. The repressive mark H3K27me3 is first deposited over the whole paternal pronucleus, in contrast to mammals where H3K27me3-mediated imprinting impacts only a handful of loci and is deposited in the female gamete (*Inoue et al., 2017*). Interference of maternal PRC2-mediated H3K27me3 deposition ultimately halts embryonic development. Thus, PRC2 initiates and maintains silencing of the entire paternal genome, which might be essential for the development of the diploid embryo of *Marchantia*.

The bryophyte life cycle marks the transition between the haploid life cycle of their aquatic ancestors and the diploid life cycle of vascular plants. Our results suggest that maintaining the dominant haploid dosage of gene expression was selected in the diploid embryo of bryophytes. In response to whole genome duplications, plant and animal genomes modulate gene expression to pre-duplication levels (*McElroy et al., 2017*; *Pala et al., 2008*; *Song et al., 2020*), though the mechanisms underlying such gene dosage control are poorly understood. Why gene dosage control in *Marchantia* was achieved through imprinting might be explained by the fact that ancestor of bryophytes had separate sexes (*Iwasaki et al., 2021*) and embryos developed on mothers. It is thus likely that viviparity, the development of embryos within mothers, and its collateral maternal support of embryo development, favoured the evolution of imprinting as a way to impose maternal control, as proposed by a body of theoretical work (*Carey et al., 2021*; *Haig, 2013*; *Haig and Wilczek, 2006*; *Montgomery and Berger, 2021*; *Shaw et al., 2011*). Imprinting has not been discovered in viviparous non-therian animals, but only orthologous loci imprinted in mammals were investigated, thus a genome-wide search may yield new insights (*Griffith et al., 2016*; *Lawton et al., 2005*; *Renfree et al., 2013*). Our findings support the idea that viviparity is sufficient for the evolution of imprinting and provide a new framework to explore the evolution of imprinting in a much more diverse range of organisms than previously considered across eukaryotes.

We believe that PCR represents a new form of imprinting. It is distinct from parental genomic imprinting described in flowering plants and therian mammals for three main reasons: the epigenetic mark is deposited after fertilization; all chromosomes of one parent are silenced by the epigenetic mark; and imprinting imposes a global maternal control of embryogenesis. The outcome of imprinting in *Marchantia* also differs from the elimination of the paternal genome following heterochromatin formation in several insect species (*Crouse, 1960*; *de la Filia et al., 2021*). In insects, heterochromatinized paternal chromosomes are eliminated by still unknown mechanisms, though the timing

of elimination varies amongst species leading to pseudohaploidy in some cases (*Bain et al., 2021*; *Morse and Normark, 2005*). In contrast, the paternal genome in *Marchantia* embryos subjected to PCR is reactivated and still passed on to the next generation. While it is possible the phenomenon we observed is due to Tak-1-specific repression, our crosses with a different male accession, Cam-1, displayed similar H3K27me3 heterochromatic domains, thus we believe PCR is parent-of-origin dependent.

PCR in *Marchantia* differs from XCI in mammals (*Heard et al., 2001*) because all paternal chromosomes are compacted and repressed and H3K9me is not involved. The formation of heterochromatic foci associated with H3K27me3 is reminiscent of the compaction and compartmentalization of the X chromosome during XCI in mammals (*Galupa and Heard, 2018*; *Nozawa et al., 2013*; *Plath et al., 2003*) or meiotic sex chromosome inactivation (*Lee and Bartolomei, 2013*; *Namekawa et al., 2006*). However, since half of the genome is marked exclusively by H3K27me3, its partition results in large compartments, the compaction of which depends on PRC2 activity. Similar mechanisms may be at play in mediating whole-chromosome compaction and repression in mammals, insects, and *Marchantia*. The precise molecular mechanisms underlying the establishment and abolition of PCR are not addressed by our model (*Figure 6*), however their elucidation will be of interest to make cross-kingdom comparisons with other instances of imprinted dosage compensation mechanisms.

Overall, we have uncovered a distinct mechanism that controls gene dosage in a haploid-dominant species. We anticipate similar controls for all bryophytes as well as other groups of organisms that alternate long-lived haploid and diploid phases. Remnants of such a control might exist in flowering plants, as suggested by maternally dominant expression in the rice zygote (*Anderson et al., 2017*). Yet, both parental genomes in *Arabidopsis* are equally expressed after fertilization (*Schon and Nodine, 2017*), but whether the total dosage of expression is the same as in haploid progenitors of gametes remains unknown. Various forms of alternation between multicellular haploid and diploid life phases are also widespread in brown and red algae. Brown algae show changes in epigenetic marks and transcription between haploid and diploid generations, despite the absence of Polycomb (*Bourdareau et al., 2021*). Thus, it would be of interest to determine the mechanisms of gene dosage control in these species as it would be distinct from PCR in *Marchantia*. Broadly, we propose that although sex chromosomes provide an important paradigm to understand gene dosage control, this phenomenon evolved several times as life cycles alternating between ploidy levels diversified, suggesting that there is an expanse of gene dosage regulatory mechanisms that remains to be explored across the broad assortment of eukaryotic life cycles.

## Materials and methods

### Plant lines and growth conditions

Wild-type male Tak-1, male Cam-1, female Cam-2, and female Tak-2 accessions of *M. polymorpha* ssp. *ruderalis* were used in this study. Cam-2 *e(z)2/e(z)3* mutants were generated as described below.

Female plants for crosses were grown at room temperature on Grodan Vital (Grodan, Roermond, The Netherlands) supplemented with liquid Hyponex fertilizer (Hyponex, Osaka, Japan) under constant white light. Male plants for crosses were grown at 22°C on Neuhaus N3 substrate soil (Humko, Podnart, Slovenia) under 16 hr of far-red light and 80% humidity. Plants grown for the collection or observation of vegetative tissues were grown under axenic conditions on half-strength Gamborg B5 media without vitamins (Duchefa Biochemie, Haarlem, The Netherlands) and 1% (w/v) agar under constant white light.

Crosses were performed by collecting mature antheridiophore discs in water and pipetting the water containing released sperm onto archegoniophores.

### Generation of e(z)2/e(z)3 mutants

To construct a plasmid to disrupt *E(z)2* and *E(z)3* simultaneously, two oligonucleotide pairs (TH219: ctcgAAATAGAAAGTGGCGCCT/TH220: aaacAGGCGCCACTTTCTATTT for *E(z)2*; TH223: ctcgATCATATACCCTCGGCTC /TH224: aaacGAGCCGAGGGTATATGAT for *E(z)3*) were annealed and cloned into the BsaI sites of pMpGE_En04 and pBC-GE14 to yield pMpGE_En04-MpEz2-sg1 and pBC-GE14-MpEz3-sg1, respectively. These two plasmids were assembled via BglI restriction sites and ligated to yield pMpGE_En04-MpEz2-sg1-MpEz3-sg1. The resulting DNA fragment containing two

MpU6promoter-gRNA cassettes was transferred into pMpGE010 (cat. no. 71536, Addgene) (*Sugano et al., 2018*) using the Gateway LR reaction (Thermo Fisher Scientific Inc, Waltham, MA) to yield pMpGE010_MpEz2-sg1-MpEz3-sg1. This construct was introduced into Cam-2 gemmae using the G-AgarTrap method (*Tsuboyama et al., 2018*). Transformants were selected for on 0.5 Gamborg B5 plates without vitamins (Duchefa Biochemie) supplemented with hygromycin and genotyped using the following primer pairs: TH300: TACGCCCTCTCCCATTGAAC/TH301: GATACGAAGAGAACGA ACCTGC for *E(z)2* and TH306: TGAGCTACATGGCTACTCTCAACC/TH307: AGCTTGGAACACGGAT CTCCTG for *E(z)3*.

## Transcriptome generation

Vegetative samples from Cam-2 and Tak-1 were collected from 100 mg of apical notches from 14-day-old plants grown from gemmae. The tissue was frozen in liquid nitrogen in Precellys tubes (Bertin Instruments, Montigny-le-Bretonneux, France) with 2.8 mm stainless steel beads (Bertin Corp., Rockville, MD) and disrupted with a Precellys Evolution tissue homogenizer (Bertin Instruments) using the following settings: 7200RPM 10 s, 5 s pause, repeated thrice. RNA was extracted using a Spectrum Plant Total RNA kit (Sigma-Aldrich, Merck KGaA, Darmstadt, Germany).

Embryo samples were collected by hand dissection, with one embryo per replicate (*Figure 1B*, *Video 1*). Embryos and the surrounding maternal calyptra tissue were dissected out of the archegoniophore into 10% RNALater (Qiagen, Hilden, Germany) on microscope slides with cavities (Marienfeld Superior, Lauda-Königshofen, Germany) and the embryo was further dissected out of the surrounding maternal tissue. Each embryo was washed four times in a series of wells containing 150 μL 10% RNALater to remove any maternal RNAs, as previously described for the pure isolation of plant embryos (*Kao and Nodine, 2020*). Each embryo was then placed in 30 μL 100% RNALater on ice until sample collection was completed. The solution was diluted to 10% RNALater by the addition of 270 μL RNase-free water (Zymo Research, Irvine, CA), vortexed gently, and the solution removed. Samples were either resuspended in 30 μL 100% RNALater and stored at −70°C or in 100 μL TRI reagent (Zymo Research). Samples were crushed with a micropestle and RNA was extracted using a Direct-zol RNA MicroPrep kit (Zymo Research). All RNA samples were treated to remove DNA using a DNase Treatment and Removal kit (Invitrogen, Thermo Fisher Scientific Inc, Waltham, MA).

RNA-seq libraries were generated from total RNA following the Smart-seq2 protocol (*Picelli et al., 2014*). cDNA synthesis was performed on 1 μL of total RNA. One μL of 10 μM 5'-Bio-anchored oligo dT ([Btn]AAGCAGTGGTATCAACGCAGAGTACTTTTTTTTTTTTTTTTTTTTTTTTTTTTTT TTTVN) and 1 μL 10 mM dNTPs were added to each sample and incubated at 72°C for 3 min and immediately placed on ice. Seven μL of a mastermix containing 0.5 μL SuperScript IV (Invitrogen), 0.25 μL RiboLock (ThermoFisher Scientific), 2 μL Superscript IV buffer (Invitrogen), 1 μL 100 mM DTT (Invitrogen), 2 μL 5 M Betaine (Sigma-Aldrich), 0.9 μL MgCl$_2$, 0.1 μL nuclease-free water (Invitrogen), and 1 μL 10 μM 5'-Bio TSO ([Btn]AAGCAGTGGTATCAACGCAGAGTACrGrG G+G, Exiqon) was added to each sample and the cDNA synthesis reaction took place under the following thermocycling conditions: 42°C 90 min, <50°C 2 min, 42°C 2 min>x10, 70°C 15 min. Forty μL of a mastermix containing 14.5 μL nuclease-free water, 25 μL Q5 Hot Start 2× MasterMix (New England Biolabs, Ipswich, MA), and 0.5 μL 10 μM 5' Bio-ISPCR oligo ([Btn]AAGCAGTGGTAT CAACGCAGAGT) was added to each sample and the PCR pre-amplification took place under the following thermocycling conditions: 98°C 3 min, <98°C 15 s, 67°C 20 s, 72°C 6 min>x12, 72°C 5 min. Samples were cleaned up by bead purification using 1× volume of MBSPure beads (Molecular Biology Services, IMP, Vienna, Austria) and samples were eluted in 15 μL of 10 mM Tris-HCl. Five to 50 ng of each sample was used for the tagmentation reaction, containing 2.5 μL of 4× TAPS-DMF buffer and 1 μL of Tn5 (Molecular Biology Services, IMP), which was 3 min at 55°C, after which samples were immediately placed on ice. Samples were purified using a DNA Clean and Concentrator kit (Zymo Research) using the manufacturer's instructions and eluted in 10 μL of 10 mM Tris-HCl. Tagmented samples were amplified by the addition of 2.5 μL each of 10 μM barcoded forward and reverse primers (*Picelli et al., 2014*) and 15 μL Q5 2× HiFI MasterMix (New England Biolabs) using the following thermocycling conditions: 72°C 3 min, 98°C 20 s, <98°C 10 s, 63°C 30 s, 72°C 3 min>x5. Amplified samples were cleaned up by bead purification using 1× volume of MSBPure beads (Molecular Biology Services, IMP). Samples were sequenced on an Illunnia NovaSeq to generate 50 bp paired-end reads. Three biological replicates each of male (Tak-1)

and female (Cam-2) vegetative tissue, 11 of wild-type (Cam-2 × Tak-1) embryos, and 17 of mutant (Cam-2 *e(z)2/e(z)3* × Tak-1) embryos were used for subsequent analyses.

## Chromatin profiling by CUT&RUN

Embryos and the surrounding calyptra were hand-dissected from archegoniophores and placed in Galbraith buffer (45 mM $MgCl_2$-6H2O, 30 mM trisodium citrate, 20 mM MOPS) pH 7.0 plus 0.1% Triton X-100 and 1× cOmplete Protease Inhibitor Cocktail (Roche, Mannheim, Germany) on ice. Samples were crushed using a mortar and pestle on ice to release nuclei and were filtered through a 40 µm filter (VWR, Radnor, PA) before staining with 2 µg/mL DAPI. Nuclei were sorted on a BD FACSARIA III (BD Biosciences, San Jose, CA) to discriminate diploid embryonic nuclei from haploid maternal nuclei. Samples were sorted into 100 µL of Wash buffer (20 mM HEPES pH 7.5, 150 mM NaCl, 0.5 mM Spermidine, 1× cOmplete Protease Inhibitor Cocktail, Roche) with 40,000 nuclei per replicate. Bio-Mag Plus Concanavalin A coated beads (Polysciences, Inc, Warrington, PA) were activated by mixing 10 µL per sample of ConA beads in 1.5 mL Binding buffer (20 mM HEPES-KOH pH 7.9, 10 mM KCl, 1 mM $CaCl_2$, 1 mM $MnCl_2$). The beads were placed on a magnet, liquid was removed, and the beads were resuspended in 1.5 mL Binding buffer. Liquid was again removed from the beads on a magnet and beads were resuspended in 10 µL Binding buffer per sample. Ten µL of the activated beads were added to each sorted nuclei sample and incubated at room temperature for 10 min on a rotator. Liquid was removed from the bead-bound nuclei on a magnet and samples were resuspended in 50 µL Antibody buffer (Wash buffer plus 2 mM EDTA). 0.5 µL of each antibody (H3K27me3 Millipore, Temecula, CA, #07-449 RRID:AB_310624; H3K36me3 Abcam, Cambridge, UK, ab9050 RRID:AB_306966; H3K9me1 Abcam ab9045 RRID:AB_306963; H3 Abcam ab1791 RRID:AB_302613) used was added to samples while gently vortexing and samples were incubated overnight at 4°C on a shaker. Liquid was removed from the samples on a magnet and washed twice in 1 mL Wash buffer before resuspending in 50 µL Wash buffer. 1.16 µL of 30 µg/mL pAG-MNase (Molecular Biology Service, IMP) was added to each sample with gently vortexing and placed on a shaker for 10 min at room temperature. Liquid was removed from the samples on a magnet and washed twice in 1 mL Wash buffer before resuspending in 150 µL Wash buffer. Three µL 100 mM $CaCl_2$ was added to ice-cold samples while gently vortexing and shaken at 4°C for 2 hr; 100 µL STOP buffer (340 mM NaCl, 20 mM EDTA, 4 mM EGTA, 50 µg/mL RNase A [ThermoFisher Scientific], 50 µg/mL glycogen, 10 pg/mL heterologous HEK293 DNA) was added to stop the reaction. Samples were incubated at 37°C for 10 min at 500 RPM then spun at 4°C for 5 min at 16,000 *g*. Samples were placed on a magnet and the liquid containing released DNA fragment was transferred to a new tube. 2.5 µL 10% SDS and 2.5 µL 20 mg/mL Proteinase K (ThermoFisher Scientific) was added to each sample, mixed by inversion, and incubated for 1 hr at 50°C. 250 µL buffered phenol-chloroform-isoamyl solution (25:24:1) was added to each sample, followed by vortexing and transfer to MaXtract tubes (Qiagen). Samples were spun for 5 min at 16,000 *g*. 250 µL chloroform was added and samples were spun for 5 min at 16,000 *g*. The top aqueous phase was transferred to a fresh tube containing 2 µL 2 mg/mL glycogen. 625 µL 100% EtOH was added before vortexing and chilling at −20°C overnight. DNA extraction continued with spinning for 10 min at 4°C at 20,000 *g*. The supernatant was poured off and 1 mL 100% EtOH was added to the samples before spinning again for 1 min at 4°C at 16,000 *g*. Supernatant was discarded and samples air-dried before dissolving in 50 µL 0.1× TE. A NEBNext Ultra II DNA library prep kit for Illumina (New England Biolabs) was used according to the manufacturer's instructions for sample library preparation. Samples were sequenced on either and Illumina HiSeqv4 or NovaSeq to generate 50 bp paired-end reads. Two biological replicates were used for each sample for H3K27me3, H3K36me3, H3K9me1, and H3 in wild-type (Cam-2 × Tak-1) embryos and mutant (Cam-2 *e(z)2/e(z)3* × Tak-1) embryos.

## Whole genome sequencing

Whole genome sequencing of Cam-2 was done as previously described (*Wang and Liu, 2020Iwasaki et al., 2021*). Five g of 14-day-old Cam-2 plants grown from gemmae were collected and frozen in liquid nitrogen. Samples were crushed using a mortar and pestle on ice and ground further in 25 mL PVPP buffer (50 mM Tris-HCl pH 9.5, 10 mM EDTA, 4 M NaCl, 1% CTAB, 0.5% PVPP, 1% beta-mercaptoethanol). The mixture was divided into two 50 mL Falcon tubes and incubated at 80°C for 30 min in a water bath. Samples were cooled to room temperature and 7.5 mL chloroform was added to each tube, followed by 5 mL TE-saturated phenol after mixing. Samples were spun at 20,000 *g*

for 5 min at room temperature and the upper aqueous phase was transferred to a new 50 mL tube. 1× volume of water and 4× volume of 100% EtOH were added and mixed, and samples were frozen at −70°C. Tubes were thawed and spun at 20,000 *g* at 4°C for 15 min. The supernatant was poured off, samples were spun again briefly, and the remaining supernatant pipetted off. Two mL of 1× TE was added to each tube and incubated at 60°C for 10 min without mixing. The supernatant was transferred to another tube and incubated at 60°C for 10 min without mixing. Two µL of RNaseA (ThermoFisher Scientific) was added and samples incubated at 37°C for 5 min. 500 µL was split into 2 mL tubes and 50 µL of 3 M sodium acetate pH 5.2 and 1 mL 100% EtOH were added. Samples were incubated at −20°C for 30 min and spun at 13,000 RPM for 15 min. After removing the supernatant, pellets were rinsed twice with 1 mL 70% EtOH and spun at 13,000 RPM for 5 min. Pellets were dried for 90 s at 65°C and resuspended in 1 mL 1× TE.

Library preparation was done by tagmentation. Briefly, 1 µL gDNA was mixed with 2.5 µL 4× TAPS-DMF buffer and 5 µL activated Tn5 (Molecular Biology Services, IMP). Tagmentation proceeded for 5 min at 55°C before cooling on ice. Samples were purified with a Zymo DNA Clean and Concentrator kit (Zymo Research) according to the manufacturer's instructions and eluted in 10 µL 10 mM Tris-HCl. PCR amplification was done by adding 2.5 µL each of 10 µM forward and reverse primers, plus 15 µL NEBNext 2× HiFi PCR MasterMix (New England Biolabs) and thermocycling with the following conditions: 72°C 3 min, 98°C 30 s, <98°C 10 s, 63°C 30 s, 72°C 3 min>x5. Samples were cleaned up by bead purification and sequenced on an Illumina NextSeq550 to generate 75 bp paired-end reads.

## Interphase nuclei immunofluorescence slide preparation

Sporophytes were hand-dissected from archegoniophores and placed in Galbraith buffer (45 mM MgCl$_2$-6H$_2$O, 30 mM trisodium citrate, 20 mM MOPS) pH 7.0 plus 0.1% Triton X-100 and 1× cOmplete Protease Inhibitor Cocktail (Roche) on ice. Samples were crushed in a mortar and pestle on ice and filtered through a 40 µm filter (VWR); 16% paraformaldehyde (PFA) was added to reach a final concentration of 4% PFA and incubated on ice for 20 min. Glycine was added to a final concentration of 125mM and nuclei were spotted onto glass slides and dried at room temperature for 20 min.

## Mitotic cells immunofluorescence slide preparation

Sporophytes were hand-dissected from archegoniophores and placed in 1x PBS with 0.1% Triton X-100 (PBST) and 4% PFA on ice. Samples were fixed by applying a vacuum for 15 min followed by 45 min at 4°C. Samples were washed thrice for 10 min each with PBST at 4°C with gentle shaking. Cell walls were digested by incubating samples in PBST plus 1% cellulase (Duchefa Biochemie) at 37°C for 10 min in a damp chamber. Samples were washed thrice for 10 min each with PBST at 4°C with gentle shaking. Intact tissues were placed in 10 µL PBST on a glass slide and squashed with a cover slip. Slides were dipped in liquid nitrogen and the cover slip was removed with a razor blade.

## Zygotic cells immunofluorescence slide preparation

To obtain archegonia holding synchronized zygote, fertilization timing was synchronized using an in vitro fertilization method described previously (*Hisanaga et al., 2021*). At 2 or 3 daf, archegoniophores were dissected under a Lynx EVO stereomicroscope (Vision Engineering, Woking, UK) and clusters of archegonia were collected into Fixative buffer (4% PFA, 1× PBS). Fixed tissues were then dehydrated and embedded in paraffin using the Donatello tissue processor (Diapath, Martinengo, Italy). Paraffin sectioning was done with a HM355S microtome (Microme, Walldorf, Germany) with 4 µm thickness. Slides were deparaffinized and rehydrated with a Gemini autostainer (Fisher Scientific) with the following protocol: xylene 5 min, xylene 5 min, EtOH 100% 5 min, EtOH 100% 5 min, EtOH 95% 5 min, EtOH 70% 5 min, EtOH 30% 5 min, running tap water 1 min, water. Antigen retrieval was performed by boiling slides in sodium citrate buffer pH 6.0.

## Immunostaining of slides

Immunostaining of slides was done by an InsituPro VSi staining system (Intavis, Cologne, Germany) as previously described, with minor modifications (*Borg et al., 2019*). Slides were washed for 10 min with TBS with 0.1% Tween-20 (TBST) five times then blocked with blocking buffer (1× TBS, 0.1% Tween-20, 2% bovine serum albumin (BSA), 5% normal goat serum) twice for 30 min each. One antibody per slide (H3K27me3 Millipore #07-449 RRID:AB_310624; H3K36me3 Abcam ab9050 RRID:AB_306966;

H3K9me1 Abcam ab9045 RRID:AB_306963) was diluted 1:100 and slides were incubated for 6 hr. After washing with TBST six times for 10 min each, slides were incubated with a 1:500 dilution of secondary antibody (Goat Anti-Rabbit IgG H&L, Alexa Fluor 488, ab150077 RRID:AB_2630356; Goat Anti-Mouse IgG H&L, Alexa Fluor 568, ab175473 RRID:AB_2895153; Goat Anti-Rabbit IgG H&L, Alexa Fluor 647, preadsorbed, ab150083 RRID:AB_2714032) and slides incubated for 2 hr. After eight 10 min washes with TBST, slides were dried and counterstained with 1.5 µg/mL 4',6-diamidino-2-phenylindole (DAPI) and mounted in Vectashield antifade mounting medium with DAPI (Vector Laboratories, Piedmont, Italy) before being sealed with a coverslip and nail varnish.

## Immunofluorescence image acquisition

Images were acquired with an LSM 780 scanning laser confocal microscope (Zeiss).

## Combined FISH and immunostaining method

Tissue fixation, nuclei isolation, and flow cytometry were performed as described (*Wang and Liu, 2020*).

A circular barrier was made with an ImmEdge Hydrophobic Barrier PAP Pen (Vector Laboratories) on the charged adhesion slide of a glass slide (ThermoFisher Scientific). Size of the circle was ~0.7 cm diameter and ≥0.5 cm line thickness. Slides were dried for 30 min. Twenty µL of the nuclei suspension was transferred into PCR tubes and nuclei were incubated at 65°C for 30 min within a PCR Thermal Cycler. Heat shock treated nuclei were immediately transferred to ice for 5 min. Five or 10 µL of 0.1 mg/mL RNase A (in 2× SSC buffer) was spotted into the circle drawn on the slide and mixed with 10 µL (containing at least $1 \times 10^4$ nuclei) of heat shock treated nuclei. The solution was spread within the circle barrier. Slides were incubated at 37°C in a ThermoBrite slide hybridizer (Leica Biosystems, Deer Park, IL) for 1 hr under a humid environment. At the end of this incubation, a very thin layer of solution remained on the glass slide. After incubation, the slides were treated for about 1 min each by dipping up and down till the streaks go away in an ethanol series (100%, 95%, 90%, 80%, 60%, 30% EtOH). The slides were then treated in antigen retrieval buffer (10 mM sodium citrate pH 6.0) at room temperature for 5 min and then the antigen retrieval was started by boiling the slides for 10–12 min in a microwave at 700 W. Slides were post-fixed in 4% formaldehyde solution 10 min after the slides cooled down to room temperature. After post-fixation, the slides were treated for about 1 min each by dipping up and down in ethanol series (30%, 60%, 80%, 90%, 95%, 100% EtOH). Slides were dried at room temperature for 1 hr.

The subsequent probe denaturation, hybridization, washing, and detection steps were performed according to *Bi et al., 2017*, with minor changes. Anti-Histone H3 (mono methyl K9) antibody (Abcam, ab9045 RRID:AB_306963) or Anti-trimethyl-Histone H3 (Lys27) Antibody (Millipore, #07-449 RRID:AB_310624) was diluted 1:500 in antibody buffer (5% BSA, 4× SSC, 0.2% Tween-20). Ten µL of the antibody mixture was pipetted onto the slides. The slides were incubated in a humid box at 37°C for 1 hr. After 1 hr of antibody binding, slides were washed for 5 min in a solution of 4× SSC with 0.2% Tween-20 in a foil-wrapped jar at room temperature on the shaker three times. 100 µl 1:150 Anti-rabbit Alexa Fluor 546-conjugated goat antibody (Invitrogen, AB_2534093 RRID:AB_2534093) was dropped onto the slides. The slides were incubated at 37°C for 1 hr followed by three times (5 min each) washing steps. Then, the slides were mounted with 5 µL SlowFade Diamond Antifade Mountant (Invitrogen). Slides were covered with a coverslip and sealed with nail polish. Images were acquired with an LSM 710 scanning laser confocal microscope (Zeiss, Oberkochen, Germany).

## Probe labelling for FISH

Probes were labeled according to the Nick Translation-based DNA Probe Labeling method (Roche). Tak-1 and Tak-2 genomic DNA was extracted by CTAB method (*Murray and Thompson, 1980*). For U chromosome probe and U chromosome competition probe, the Tak-2 gDNA was used as template. For V chromosome probe and V chromosome competition probe, the Tak-1 gDNA was used as template. Fluoroprobe labelling mix: For V/U chromosome probe (dATP, dCTP, dGTP, dTTP, Dig-dUTP); For V/U chromosome competition probe (dATP, dCTP, dGTP, dTTP). For U chromosome FISH, U-chromosome probe and five times V-chromosome competition probe were loaded. For V chromosome FISH, V-chromosome probe and five times U-chromosome competition probe were loaded.

## Tissue clearing and DAPI staining of zygotes

Tissue clearing and DAPI staining for 3 daf zygotes were done as described previously (*Hisanaga et al., 2021*). Stained samples were mounted in Vectashield antifade mounting medium with DAPI (Vector Laboratories). Images were taken with an LSM 780 confocal microscope (Zeiss).

## Nuclear envelope visualization

To observe the nuclear envelope of 3 daf zygotes, ECpro:SUN-GFP (*Hisanaga et al., 2021*) females were fertilized with wild-type sperm and zygotes were excised under a Lynx EVO stereomicroscope (Vision Engineering) and mounted in half-strength Gamborg B5 media without vitamins (Duchefa Biochemie) liquid medium. Samples were observed under a Nikon C2 confocal laser scanning microscope (Nikon Instech, Tokyo, Japan).

## Mutant fitness analyses

Four gemmae from Cam-2 and Cam-2 *e(z)2/e(z)3* plants were grown together. Images of each gemmaling was taken at 4, 7, and 10 days after planting using a Lynx EVO stereomicroscope (Vision Engineering). The area of each gemmaling was calculated using FIJI v2.0.0 (*Schindelin et al., 2012*) and plotted as a smoothed curve using the LOESS function and formula *y~x* in R v3.5.1 (*R Development Core Team, 2018*) with the ggplot2 v3.3.5 package (*Wickham, 2016*).

Gemmae from Cam-2 and Cam-2 *e(z)2/e(z)3* were planted on Grodan and monitored until the first archegoniophores were visible. Pictures were taken after all replicates had produced archegoniophores to illustrate the synchronicity of archegoniophore developmental stage.

Images of fully dissected embryos (see Transcriptome generation section above for details) were taken with Lynx EVO stereomicroscope (Vision Engineering). The height and width of each embryo was calculated in FIJI v2.0.0 (*Schindelin et al., 2012*) using images of a calibration slide as reference. The sample area was calculated by multiplying height and width.

Aborted embryos can be identified by a browning of tissue, collapse of tissue within the calyptra, and the outgrowth of the perianth without growth of the embryo within. Embryo survival was calculated as the number of green, non-collapsed embryos per archegoniophore divided by the number of perianths with or without live embryos.

Mature embryos can be identified by the yellowing of tissue due to the production of spores within. The percentage of embryos producing spores was calculated as the number of mature yellow embryos per archegoniophore divided by the number of perianths with or without live embryos.

Spore germination was assessed by counting the number of sporelings growing out from spots of serially diluted spore solutions from single sporophytes. Mature embryos were dissected from archegoniophores, dried for 1 week, and frozen at −70°C. Frozen embryos were thawed and ruptured in 100 µL sterile water using a sterile pipette tip. 80 µL of the spore suspension was transferred to a tube containing 420 µL sterile water. 500 µL of 0.1% NaDCC (Sigma-Aldrich) was added to each sample and tubes were inverted and spun at 13,000 RPM for 1 min. The supernatant was removed, and spores were resuspended in 100 µL sterile water. 20 µL of spore suspension was spotted onto plates of half-strength Gamborg B5 media without vitamins (Duchefa Biochemie) and 1% agar. 20 µL of spore suspension was carried to a tube containing 60 µL sterile water. The process was repeated until dilutions of 1:1, 1:4, 1:16, and 1:64 were spotted. Images of sporeling germination and growth were taken at 11 days after planting.

## Transcriptome analysis

Published transcriptomes from male and female reproductive tissues, antheridiophores, and archegoniophores, respectively *Higo et al., 2016* and wild-type Tak-2 × Tak-1 embryos (*Frank and Scanlon, 2015*) were downloaded from the SRA database.

Reads were mapped to the Takv6 genome (*Iwasaki et al., 2021*) wherein all SNP positions between Tak-1 and Cam-2 or between Tak-1 and Tak-2 were replaced with N's, depending on the genotype of the sample (refer to SNP data analysis section below). Reads were preprocessed with SAMtools v1.9 (*Li et al., 2009b*) and BEDTools v2.27.1 (*Quinlan and Hall, 2010*), trimmed with Trim Galore (https://github.com/FelixKrueger/TrimGalore, *Krueger, 2022*) and mapped with STAR v2.7.1 (*Dobin et al., 2013*). TPM values were calculated by RSEM v1.3.2 (*Li and Dewey, 2011*). Data from RSEM were imported into R v3.5.1 (*R Development Core Team, 2018*) using the tximport package v1.10.1

(*Soneson et al., 2015*). Differential gene analysis was performed using DeSeq2 v1.22.2 (*Love et al., 2014*). Principal component analysis was performed in R v3.5.1 (*R Development Core Team, 2018*). Effect size (Cohen's *d*) was calculated in R using effsize v0.7.6 (*Torchiano, 2020*) where $|d| < 0.2$ is no effect, $0.2 < |d| < 0.5$ is a small effect, $0.5 < |d| < 0.8$ is a medium effect, and $|d| > 0.8$ is a large effect, as previously reported (*Cohen, 1992*). Heatmaps were generated in R using the pheatmap v1.0.12 package (*Kolde, 2019*).

## CUT&RUN data analysis

Reads were mapped to the Takv6 genome (*Iwasaki et al., 2021*) wherein all SNP positions between Tak-1 and Cam-2 were replaced with N's (refer to SNP data analysis section below). File processing and mapping parameters were performed as previously published (*Montgomery et al., 2020*). Chromatin enrichment per gene was calculated by counting the number of reads and normalizing to 1× genome coverage.

H3K27me3 peaks were called using HOMER v4.9 (*Heinz et al., 2010*) and considered to associate with a gene if it overlapped with at least 50% of the peak length using BEDTools v2.27.1 (*Quinlan and Hall, 2010*). $\chi^2$ test was performed in R v3.5.1 (*R Development Core Team, 2018*) for downregulated genes overlapping H3K27me3 peaks in *e(z)2/e(z)3* mutant embryos (*Supplementary file 2*).

## SNP data analysis

Reads were preprocessed with SAMtools v1.9 (*Li et al., 2009b*), BEDTools v2.27.1 (*Quinlan and Hall, 2010*), and Picard v2.18.27 (http://broadinstitute.github.io/picard/, *Soifer, 2022*) before mapping to the Tak-1 genome with bwa v0.7.17 (*Li and Durbin, 2009a*). SNPs were called using gatk v4.0.1.2 and the reference genome with all SNPs replaced with N's was created (*McKenna et al., 2010*).

Mapped reads from CUT&RUN and RNA-seq experiments were assigned to paternal or maternal genomes using SNPSplit v0.3.4 (*Krueger and Andrews, 2016*). Counts for the number of reads originating from either genome were calculated per sample using SAMtools v1.9 (*Li et al., 2009b*) and BEDTools v2.27.1 (*Quinlan and Hall, 2010*). The maternal ratio was determined by dividing the number of maternal reads by total reads per gene. For CUT&RUN data, only data from genes with more than 10 total reads in each replicate were retained. For RNA-seq data, only data from genes with 50 or more reads in total across all replicates were retained. Additionally, data from genes that were not completely maternal in female Cam-2 RNA-seq data ($p_m < 0.95$) or were not completely paternal in male Tak-1 RNA-seq data ($p_m > 0.05$) were excluded from further maternal ratio analyses if there were at least five reads.

The statistical analysis of RNA-seq SNPs is based on prior analyses (*de la Filia et al., 2021*; *McDonald, 2014*). For each gene and replicate, an exact binomial test was performed against the null hypothesis of Mendelian expression ($p_m = 0.5$). In cases where the number of reads was not a discrete count, the number was rounded down before the exact test. The non-rounded values were used for estimating the expression ratio. The p-values were Bonferroni corrected for multiple testing. Significant (adjusted $p < 0.05$) cases were classed, based on the ratio estimate as maternal if $p_m \geq 0.95$, maternally biased if $0.65 \leq p_m < 0.95$, paternally biased if $0.05 < p_m \leq 0.35$ and paternal if $p_m \leq 0.05$. Cases with $0.35 < p_m < 0.65$ as well as all non-significant results were classed as unbiased. Then, each gene was tested for heterogeneity across replicates using G-tests. As in *de la Filia et al., 2021*, all genes that did not show significant heterogeneity across replicates ($p > 0.05$) were automatically included in the final analysis. Significantly heterogeneous genes were kept only if all replicates were all either significant or all non-significant in the exact binomial test and their ratios fell into the same category. For the kept genes, the paternal and maternal SNP counts from all replicates were pooled, the expression ratios were estimated, and a final exact binomial test was performed. The expression ratios from the pooled data were used to categorize the genes into their final groups (maternal: $p_m \geq 0.95$, maternally biased: $0.65 \leq p_m < 0.95$, unbiased: $0.35 < p_m < 0.65$, paternally biased: $0.05 < p_m \leq 0.35$, and paternal: $p_m \leq 0.05$). The Bonferroni corrected p-values were used to call significance of the non-Mendelian expression.

The classification and significance testing for CUT&RUN data were done the same way as for the RNA-seq data, with the one exception that the final expression ratios were estimated as the mean of the expression ratios estimated in the two replicates separately.

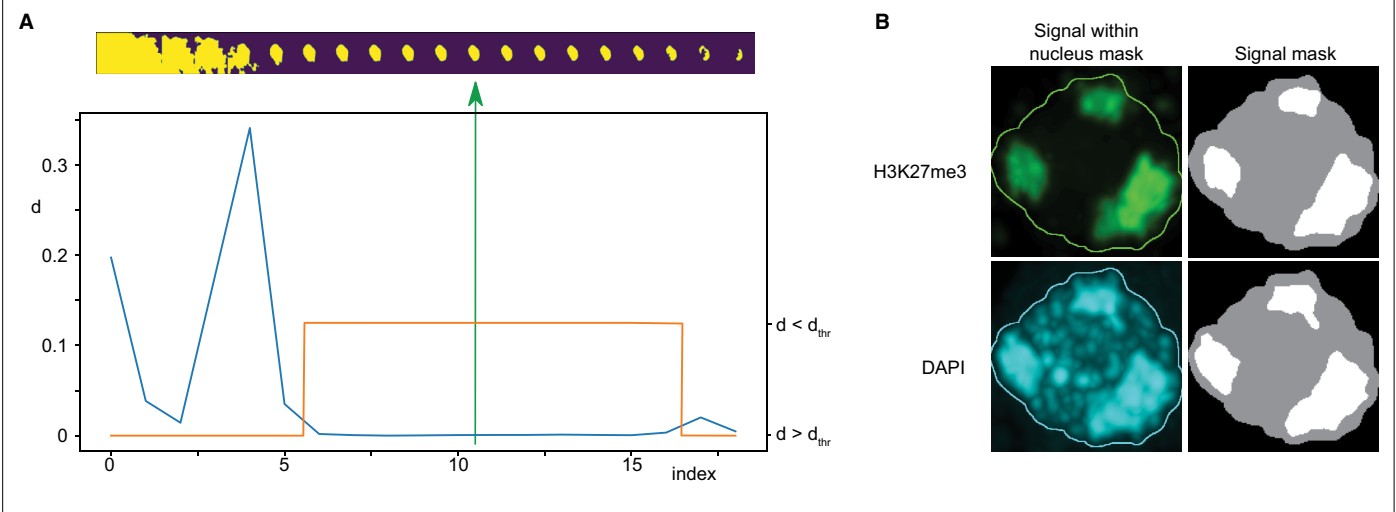

**Figure 7.** Immunofluorescence image analysis. (**A**) Nucleus segmentation. Top panel: A sequence of 20 segmentations created by thresholding. Bottom panel: Differences (*d*) between nucleus mask sizes (blue curve). Plateau identified by estimating a threshold value $d_{thr}$ (orange curve). The selected segmentation is in the centre of the plateau (green arrow). (**B**) Foci segmentation. Left: H3K27me3 and DAPI image bands with the overlayed nucleus border detected from the DAPI band. Right: Masks of the nucleus and detected foci.

## Interphase nuclei image deconvolution

Immunofluorescence images of interphase nuclei were deconvolved with Huygens Professional v21.04 (Scientific Volume Imaging BV, Hilversum, The Netherlands) using the CMLE algorithm with 40 iterations and SNR values as follows: 6 for WT H3K27me3 samples H3K27me3 channel, 8 for WT H3K27me3 samples DAPI channel, 2 for WT H3K9me1 samples H3K9me1 channel, 4 for WT H3K9me1 samples DAPI channel, 4 for WT H3K36me3 samples H3K36me3 channel, 5 for WT H3K36me3 samples DAPI channel, 2 or 4 for mutant H3K27me3 samples H3K27me3 channel, 3 for mutant H3K27me3 samples DAPI channel.

## Interphase nuclei image nuclei segmentation

Nuclei were identified from DAPI signal marking DNA in each immunofluorescence image. An adaptive thresholding technique was used, based on the creation of a sequence of 20 threshold values spanning a range from a clearly too low threshold to a clearly too high threshold. A sequence of masks was thus obtained for each three-dimensional image by thresholding it using these values. Subsequently, a maximum intensity projection of each mask was computed and size of each mask projection was evaluated. Typically, starting from the lowest threshold, such sequence first decreased rapidly, followed by a wide plateau, and ending by a decreasing tail near the highest threshold value (*Figure 7A*). The nearly constant plateau was detected by thresholding absolute value of neighbour size differences. Again, a sequence of thresholds was used, starting from a minimum equal to 1/10 of average of the differences, until the length of such plateau was larger than 6. Finally, the segmentation in the middle of the plateau was taken as the final one. In the last step eventual holes in the three-dimensional mask were filled by a hole-filling operation and eventual thin gaps in the mask were filled by binary closing.

Nucleus masks thus obtained were subsequently visually inspected. Except for a few cases, the nuclei were labeled correctly. The incorrect ones were manually adjusted by means of drawing functions in FIJI v2.0.0 (*Schindelin et al., 2012*).

## Interphase nuclei image foci segmentation

Foci within the nucleus area were detected separately for the DAPI and the immunofluorescence channels by *k*-means classification in 2, 3, and 4 classes, which in our case of single-valued data specified one, two, or three threshold values. The foci mask was then computed by thresholding the input data using the highest of these thresholds. The results were visually inspected and classification

in three classes was then taken for further processing and evaluation. Only the largest foci with size bigger than 20% of the nucleus size were considered (*Figure 7B*).

In a limited number of cases, one or two foci were missing, or foci were too large. In the first case there were two possibilities of how to identify more foci: either by decreasing the expected foci size or by decreasing the threshold value. Thus, in a loop we multiplied both these values by a coefficient until the desired number of foci was reached. In the second case with too large foci, classification in four classes was used, which increased the highest threshold and simultaneously decreased the size of the foci.

## Statistical analyses

Statistical comparisons means of FISH immunostaining images and mutant growth analyses were performed with Wilcoxon tests in R v3.5.1 (*R Development Core Team, 2018*) with the ggpubr v0.4.0 package (*Kassambara, 2020*). Spearman correlations were calculated in R and with the ggpubr package.

## Acknowledgements

We thank A Pauli, A Burga, and I Patten for suggestions and critical reading of the manuscript. FB acknowledges support from the PlantS, Next Generation Sequencing and histopathology facilities at the Vienna BioCenter Core Facilities (VBCF), and the BioOptics facility and Molecular Biology Services from the Institute for Molecular Pathology (IMP), members of the Vienna BioCenter (VBC), Austria. This work was funded by FWF grants I4258, P26887, P28320, P32054, P33380 to FB, FWF doctoral school DK W1238 to SAM, and European Research Council under the European Union's Horizon 2020 research and innovation programme 757600 to CL. This project has received funding from the European Union's Framework Programme for Research and Innovation Horizon 2020 (2014–2020) under the Marie Curie Skłodowska Grant Agreement No. 847548.

## Additional information

### Funding

| Funder | Grant reference number | Author |
| --- | --- | --- |
| Austrian Science Fund | P26887 | Frédéric Berger |
| Austrian Science Fund | P28320 | Frédéric Berger |
| Austrian Science Fund | P32054 | Frédéric Berger |
| Austrian Science Fund | P33380 | Frédéric Berger |
| Austrian Science Fund | W1238 | Sean Akira Montgomery |
| H2020 European Research Council | 757600 | Chang Liu |
| Horizon 2020 Framework Programme | 847548 | Tetsuya Hisanaga |
| Austrian Science Fund | I4258 | Frédéric Berger |

The funders had no role in study design, data collection and interpretation, or the decision to submit the work for publication.

### Author contributions

Sean Akira Montgomery, Conceptualization, Resources, Data curation, Software, Formal analysis, Validation, Investigation, Visualization, Methodology, Writing – original draft, Writing – review and editing; Tetsuya Hisanaga, Resources, Investigation, Methodology; Nan Wang, Investigation, Methodology; Elin Axelsson, Resources, Data curation, Software, Formal analysis; Svetlana Akimcheva, Resources; Milos Sramek, Resources, Software, Formal analysis, Methodology; Chang Liu, Supervision, Funding acquisition, Writing – review and editing; Frédéric Berger, Conceptualization,

Supervision, Funding acquisition, Writing – original draft, Project administration, Writing – review and editing

## Author ORCIDs
Sean Akira Montgomery http://orcid.org/0000-0003-1680-4858
Tetsuya Hisanaga http://orcid.org/0000-0002-2834-7044
Elin Axelsson http://orcid.org/0000-0003-4382-1880
Frédéric Berger http://orcid.org/0000-0002-3609-8260

## Decision letter and Author response
Decision letter https://doi.org/10.7554/eLife.79258.sa1
Author response https://doi.org/10.7554/eLife.79258.sa2

---

# Additional files

## Supplementary files
• Supplementary file 1. List of *Marchantia* chromatin-related genes and their expression status in embryos relative to other tissues.

• Supplementary file 2. Data for $\chi^2$ test. List of all *Marchantia* genes and the presence or absence of H3K27me3 peaks in *e(z)2/e(z)3* mutant embryos plus whether the genes were upregulated, downregulated, or unchanged in a differential gene expression analysis between wild-type and *e(z)2/e(z)3* mutant embryos.

• MDAR checklist

## Data availability
The CUT&RUN and RNA-seq sequencing datasets generated for the current study is available in the Gene Expression Omnibus (GEO) under series GSE193307. Whole-genome sequencing data are deposited under BioProject accession number PRJNA795113. Publicly available datasets can be accessed under the DDBJ Sequence Read Archive accession numbers DRR050346-DRR050348 and DRR050351-DRR050353 and the NCBI Sequence Read Archive accession numbers SRR1553297-SRR1553299. Source data are provided with this paper. Original images are deposited online at FigShare. https://doi.org/10.6084/m9.figshare.19249622 and https://doi.org/10.6084/m9.figshare.19249643. All original code has been deposited online at FigShare. https://doi.org/10.6084/m9.figshare.19249592.

The following datasets were generated:

| Author(s) | Year | Dataset title | Dataset URL | Database and Identifier |
|---|---|---|---|---|
| Montgomery SA, Hisanaga T, Wang N, Axelsson E, Akimcheva S, Sramek M, Liu C, Berger F | 2022 | Whole-genome sequencing data | https://www.ncbi.nlm.nih.gov/bioproject/?term=PRJNA795113 | NCBI BioProject, PRJNA795113 |
| Montgomery SA, Hisanaga T, Wang N, Axelsson E, Akimcheva S, Sramek M, Liu C, Berger F | 2022 | Confocal microscopy images | https://doi.org/10.6084/m9.figshare.19249622 | figShare, 10.6084/m9.figshare.19249622 |
| Montgomery SA, Hisanaga T, Wang N, Axelsson E, Akimcheva S, Sramek M, Liu C, Berger F | 2022 | Plant images for size and fitness measurements | https://doi.org/10.6084/m9.figshare.19249643 | figshare, 10.6084/m9.figshare.19249643 |
| Montgomery SA, Hisanaga T, Wang N, Axelsson E, Akimcheva S, Sramek M, Liu C, Berger F | 2022 | Original code for Polycomb-mediated repression of paternal chromosomes maintains haploid dosage in diploid embryos of Marchantia | https://doi.org/10.6084/m9.figshare.19249592 | figshare, 10.6084/m9.figshare.19249592 |

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
