## [Editor Report]

Mechanisms for controlling gene dosage and uniparental gene expression vary widely across the eukaryotic tree, with many such mechanisms still unknown. This article describes an epigenetic mechanism used to modulate paternal chromosome gene dosage during the transient diploid state of the primarily haploid plant, Marchantia polymorpha. This fascinating case of genome-wide genomic imprinting will be of broad interest to scientists studying gene expression, early development, and especially those focused on understanding the diversity of gene dosage control mechanisms.

---

## [Decision Letter]

**Decision letter after peer review:**

Thank you for submitting your article "Polycomb-mediated repression of paternal chromosomes maintains haploid dosage in diploid embryos of Marchantia" for consideration by *eLife*. Your article has been reviewed by 3 peer reviewers, and the evaluation has been overseen by a Reviewing Editor and Jessica Tyler as the Senior Editor. The reviewers have opted to remain anonymous.

Essential revisions:

1) Given that Tak-1 is the male in both the Cam-2 x Tak-1 and the Tak-2 x Tak-1 experiments, one can't rule out the possibility that the paternal chromosome silencing, H3K27me enrichment, etc. is specific to Tak-1. Under this alternative model, Tak-1 chromosomes are silenced in combination with either Cam-2 or Tak-2 – a phenomenon more akin to nucleolar dominance – i.e. genotype-dependent rather than parent-of-origin dependent. Assessing biased expression of diploid embryos from a cross using a different male genotype would blunt this concern. If such a cross is not possible, then addressing this alternative model in the Discussion is warranted.

2) Please include statistical tests for deviations from the expected 1:1 maternal:paternal ratio (see various imprinting studies).

3) The authors show that e(z) mutant embryos grow more slowly and that most do not survive. They also show that mutants have reduced maternal allele bias. In terms of linking the phenotype to the gene expression change, it would be important to show that the total expression level of individual genes was altered in the mutants (for example, increased paternal allele expression might be compensated for by decreased maternal allele expression, in which case it would harder to connect the mutant phenotype to PCI). The authors should evaluate how many genes are differentially expressed between wild-type and mutant embryos.

4) The proposed post-fertilization mechanism of imprinting is exciting, but this part of the manuscript seems somewhat underdeveloped. Figure 4 shows the enrichment of H3K27Me3 in paternal pronuclei at the daf 3 stage, but it would help to see 1) the earlier stages at which this signal is missing; and 2) staining for other histone modifications, to verify that the enrichment is really specific to H3K27Me3.

Additional revisions requested:

5) It's satisfying that when ez2 and ez3 are disrupted (Figure 5, Figure 5-Figure supp 1D), that the IF for mutant embryos looks like H3K27me3 in vegetative nuclei (Figure 3A). But the paternal bias of H3K27me3 is still quite prevalent (Figure 5-Figure supp1F) as is the maternal bias in transcription – the transcriptional ratio is not close to 50:50 (Figure 5B). The authors should comment or speculate on how paternal bias of H3K27me3 persists in this mutant, given their model. Perhaps the remaining H3K27me3 is from paternally supplied E(z). Since the paternal and maternal pronuclei are segregated for quite some time, a paternally supplied factor could also specifically mark one chromosome set (although it is less clear why this would be so from an evolutionary perspective). Generating paternal E(z) mutants would be interesting, but is likely beyond the current scope.

6) From the genetic results, one can conclude that E(z)2 and E(z)3 are essential for viability and fecundity. But it is not yet clear, as claimed on line 339, that PCI is essential for viability and fecundity, as E(z)2 and E(z)3 may also have roles beyond or in addition to PCI. I suggest dividing this sentence into what one can conclude from the genetics, and what this suggests about the possible importance of PCI.

7) Paternal chromosome inactivation is perhaps too strong of a phrase to describe this very interesting phenomenon. There are thousands of genes for which expression is biased toward the maternal allele, but detectable paternal allele transcript is present in the embryos. It is important to get the name right now, because it may influence the field for a long time (consider the many caveats now required for the term "inactive X") The reviewers came up with some alternative terms, "PCR" (Paternal Chromosome Repression) or "PCS" (Paternal Chromosome Suppression/Silencing). This change is up to the author's discretion.

8) At 3 days after fertilization both H3 and H3K27me3 are present, but since histones replaced protamines, the critical events of histone deposition and Polycomb marking might have occurred simultaneously or more likely successively. Can the authors distinguish these possibilities?

9) Comparisons are made to X-chromosome inactivation in mammals and male X upregulation in *Drosophila* as silencing phenomena that are not conserved in evolution, but there are others that might be more relevant, such as paternal genome elimination in mealybugs, which is a spermatocyte-specific event that follows whole genome silencing during embryogenesis. Another is Meiotic Sex Chromosome Inactivation, a conserved phenomenon in animals that targets unpaired chromosomes.

10) Low outliers in Figure 1-Figure suppl 1F (maternal transcription bias) and high outliers in Figure 2 suppl 1E (paternal H3K27me3 bias). Are they the same? Are they escapers?

11) Can the authors provide additional evidence that the large heterochromatic H3K27Me3-stained foci (Figure 3) do represent or at least contain paternal chromosomes? Although it's certainly suggested by the images of mitotic chromosomes in Figure 3B, more direct evidence would strengthen this point. Ideally, DNA FISH for a region that can distinguish between the strains can be used. But if no such region is available, DNA FISH against common loci or chromosomal regions would be expected to show a spot within the heterochromatic foci (paternal) and a spot outside of them (maternal), which would corroborate the idea that paternal chromosomes are contained within the visible foci. Combining or supplementing this with RNA FISH would be even better, but could be technically challenging.

Related to the point above, it would further strengthen the proposed model if the authors could demonstrate the localization of the embryo-specific E(z) homologues to the daf3 paternal pronucleus and their absence from the maternal pronucleus. If this experiment is not possible, addressing this limitation in the Discussion is warranted.

12) Figure 4 – supplement 1 shows many images of what appears to be the same type of staining, and the purpose behind this is unclear. Are these all the images used in quantification, are they just representative images or are they supposed to display certain variations within this IF?

13) Please define "viviparity" and its relevance for the broader audience.

14) The font size on graphs within main figures should be made uniform and probably increased too.

---

## [Author Response]

Essential revisions:1) Given that Tak-1 is the male in both the Cam-2 x Tak-1 and the Tak-2 x Tak-1 experiments, one can't rule out the possibility that the paternal chromosome silencing, H3K27me enrichment, etc. is specific to Tak-1. Under this alternative model, Tak-1 chromosomes are silenced in combination with either Cam-2 or Tak-2 – a phenomenon more akin to nucleolar dominance – i.e. genotype-dependent rather than parent-of-origin dependent. Assessing biased expression of diploid embryos from a cross using a different male genotype would blunt this concern. If such a cross is not possible, then addressing this alternative model in the Discussion is warranted.

We thank the reviewer for their comments. The only male Marchantia accession with a sequenced genome is Tak-1 so we opted not to sequence and assemble the genome of another male accession. However, we were able to perform a cross between Tak-2 and a different male accession, Cam-1, and observed immunostaining of embryonic nuclei. We observe the same condensed heterochromatic domains tightly associated with H3K27me3 in Tak-2 x Cam-1 crosses and suggest that is it likely that the same phenomenon is at play. These data have been incorporated into Figure 3—figure supplement 1 and comments lines 204-207. In the Discussion we have included the alternative hypothesis the reviewer proposed at lines 425-427.

2) Please include statistical tests for deviations from the expected 1:1 maternal:paternal ratio (see various imprinting studies).

We thank the reviewer for this comment. We have performed statistical tests for maternal bias for transcription and H3K27me3 in WT and mutant and indicated the number of genes that are significant in each of our five categories of parental bias (Paternal, Paternal bias, No bias, Maternal bias, Maternal) in Figures 1 and 2 and Figure 1—figure supplement 1, Figure 2—figure supplement 1, Figure 5—figure supplement 1 and 2 and lines 166-168.

3) The authors show that e(z) mutant embryos grow more slowly and that most do not survive. They also show that mutants have reduced maternal allele bias. In terms of linking the phenotype to the gene expression change, it would be important to show that the total expression level of individual genes was altered in the mutants (for example, increased paternal allele expression might be compensated for by decreased maternal allele expression, in which case it would harder to connect the mutant phenotype to PCI). The authors should evaluate how many genes are differentially expressed between wild-type and mutant embryos.

We thank the reviewer for this comment. We have now indicated the number of differentially expressed genes there are between wild-type and mutant embryos at lines 319-320.

4) The proposed post-fertilization mechanism of imprinting is exciting, but this part of the manuscript seems somewhat underdeveloped. Figure 4 shows the enrichment of H3K27Me3 in paternal pronuclei at the daf 3 stage, but it would help to see 1) the earlier stages at which this signal is missing; and 2) staining for other histone modifications, to verify that the enrichment is really specific to H3K27Me3.

We thank the reviewer for this suggestion. We have now included H3K27me3 immunofluorescence experiments at 2 daf to show a lack of this mark at this stage in Figure 4—figure supplement 1 (line 269-271). We have also included H3K9me1 IF experiments at 2 daf and 3 daf and show that a weak signal of H3K9me1 also appears in the paternal pronucleus at 3 daf (line 276-277). We do not make comparisons to the presence or absence of histone modifications in the maternal pronucleus because its large size hampers capturing images from the entire maternal pronucleus within a single paraffin slice (line 277-279, 1249-1255).

Additional revisions requested:5) It's satisfying that when ez2 and ez3 are disrupted (Figure 5, Figure 5-Figure supp 1D), that the IF for mutant embryos looks like H3K27me3 in vegetative nuclei (Figure 3A). But the paternal bias of H3K27me3 is still quite prevalent (Figure 5-Figure supp1F) as is the maternal bias in transcription – the transcriptional ratio is not close to 50:50 (Figure 5B). The authors should comment or speculate on how paternal bias of H3K27me3 persists in this mutant, given their model. Perhaps the remaining H3K27me3 is from paternally supplied E(z). Since the paternal and maternal pronuclei are segregated for quite some time, a paternally supplied factor could also specifically mark one chromosome set (although it is less clear why this would be so from an evolutionary perspective). Generating paternal E(z) mutants would be interesting, but is likely beyond the current scope.

We thank the reviewer for these comments. We state on line 315-317 that maternal E(z)1 is expressed in embryos and may account for the remaining H3K27me3. We have included a sentence that paternal E(z)2 and E(z)3 may also account for the remaining H3K27me3 in lines 318-319.

6) From the genetic results, one can conclude that E(z)2 and E(z)3 are essential for viability and fecundity. But it is not yet clear, as claimed on line 339, that PCI is essential for viability and fecundity, as E(z)2 and E(z)3 may also have roles beyond or in addition to PCI. I suggest dividing this sentence into what one can conclude from the genetics, and what this suggests about the possible importance of PCI.

We are sorry that the reviewer found the sentence confusing. We have made a longer, more explicit statement on lines 357-359.

7) Paternal chromosome inactivation is perhaps too strong of a phrase to describe this very interesting phenomenon. There are thousands of genes for which expression is biased toward the maternal allele, but detectable paternal allele transcript is present in the embryos. It is important to get the name right now, because it may influence the field for a long time (consider the many caveats now required for the term "inactive X") The reviewers came up with some alternative terms, "PCR" (Paternal Chromosome Repression) or "PCS" (Paternal Chromosome Suppression/Silencing). This change is up to the author's discretion.

We thank the reviewers for their suggestions and have opted to change the name to PCR (Paternal Chromosome Repression).

8) At 3 days after fertilization both H3 and H3K27me3 are present, but since histones replaced protamines, the critical events of histone deposition and Polycomb marking might have occurred simultaneously or more likely successively. Can the authors distinguish these possibilities?

We thank the reviewers for their questions. We have now included immunostaining images at 2 days after fertilization and show that H3 is present whereas H3K27me3 is absent in Figure 4—figure supplement 1 (line 269-271). This suggests that H3K27me3 deposition occurs after histone deposition, but cannot exclude an earlier deposition of low levels of H3K27me3 (275-276).

9) Comparisons are made to X-chromosome inactivation in mammals and male X upregulation in *Drosophila* as silencing phenomena that are not conserved in evolution, but there are others that might be more relevant, such as paternal genome elimination in mealybugs, which is a spermatocyte-specific event that follows whole genome silencing during embryogenesis. Another is Meiotic Sex Chromosome Inactivation, a conserved phenomenon in animals that targets unpaired chromosomes.

We thank the reviewer for their comment. We have now made reference to both MSCI and mealybug PGE in our Discussion on lines 432-433 and 419-420.

10) Low outliers in Figure 1-Figure suppl 1F (maternal transcription bias) and high outliers in Figure 2 suppl 1E (paternal H3K27me3 bias). Are they the same? Are they escapers?

We thank the reviewer for this question. In Figure 2D, we plot transcription bias against H3K27me3 bias, and no genes are present in the lower right quadrant, which would represent the low outliers in Figure 1-Figure suppl 1F and high outliers in Figure 2 suppl 1E. Therefore, they are not the same, but could potentially be escapers.

11) Can the authors provide additional evidence that the large heterochromatic H3K27Me3-stained foci (Figure 3) do represent or at least contain paternal chromosomes? Although it's certainly suggested by the images of mitotic chromosomes in Figure 3B, more direct evidence would strengthen this point. Ideally, DNA FISH for a region that can distinguish between the strains can be used. But if no such region is available, DNA FISH against common loci or chromosomal regions would be expected to show a spot within the heterochromatic foci (paternal) and a spot outside of them (maternal), which would corroborate the idea that paternal chromosomes are contained within the visible foci. Combining or supplementing this with RNA FISH would be even better, but could be technically challenging.

What is suggested is technically challenging since it requires the development of allele specific oligopaints for entire sets of chromosomes. This would be an article in its own self and therefore beyond the scope of revision. The fact that exactly 8 autosomes out of 16 are covered by H3K27me3 and that the ChIP-seq data shows that H3K27me3 covers thousands of paternal alleles scattered along all autosomes demonstrate that the eight chromosomes covered with H3K27me3 in metaphase plates are the paternal chromosomes.

Related to the point above, it would further strengthen the proposed model if the authors could demonstrate the localization of the embryo-specific E(z) homologues to the daf3 paternal pronucleus and their absence from the maternal pronucleus. If this experiment is not possible, addressing this limitation in the Discussion is warranted.

This would be an interesting point but it would require the development of several new transgenic lines and is the departure point of another story about the mechanism that targets the PRC2 complex to the male pronucleus. Adding such a piece of information would not add to the main conclusion of this manuscript dedicated entirely to document the extraordinary fact that the paternal chromosomes are imprinted.

12) Figure 4 – supplement 1 shows many images of what appears to be the same type of staining, and the purpose behind this is unclear. Are these all the images used in quantification, are they just representative images or are they supposed to display certain variations within this IF?

The images in Figure 4 – supplement 1 are replicates to show the consistency of the observed result. The number of images has been reduced to illustrate representative images and this has been noted in the figure legend.

13) Please define "viviparity" and its relevance for the broader audience.

This has been corrected on lines 405-406.

14) The font size on graphs within main figures should be made uniform and probably increased too.

This has been corrected.